# Do Different Tracking Tasks Require Different Appearance Models?

**Zhongdao Wang**[1,2]  **Hengshuang Zhao**[3,4]  **Ya-Li Li**[1,2]

**Shengjin Wang**[1,2*]  **Philip H.S. Torr**[3]  **Luca Bertinetto**[5]

[1]Beijing National Research Center for Information Science and Technology (BNRist)
[2]Department of Electronic Engineering, Tsinghua University
[3]Torr Vision Group, University of Oxford
[4]The University of Hong Kong
[5]Five AI

https://zhongdao.github.io/UniTrack

## Abstract

Tracking objects of interest in a video is one of the most popular and widely applicable problems in computer vision. However, with the years, a Cambrian explosion of use cases and benchmarks has fragmented the problem in a multitude of different experimental setups. As a consequence, the literature has fragmented too, and now novel approaches proposed by the community are usually specialised to fit only one specific setup. To understand to what extent this specialisation is necessary, in this work we present **UniTrack**, a solution to address *five* different tasks within the same framework. UniTrack consists of a single and task-agnostic appearance model, which can be learned in a supervised or self-supervised fashion, and multiple "heads" that address individual tasks and do not require training. We show how most tracking tasks can be solved within this framework, and that the *same* appearance model can be successfully used to obtain results that are competitive against specialised methods for most of the tasks considered. The framework also allows us to analyse appearance models obtained with the most recent self-supervised methods, thus extending their evaluation and comparison to a larger variety of important problems.

## 1 Introduction

Unlike popular image-based computer vision tasks such as classification and object detection, which are (for the most part) unambiguous and clearly defined, the problem of *object tracking* has been considered under different setups and scenarios, each motivating the design of a separate set of benchmarks and methods. For instance, for the Single Object Tracking (SOT) and Video Object Segmentation (VOS) communities [70, 29, 48], *tracking* means estimating the location of an arbitrary user-annotated target object throughout a video, where the location of the object is represented by a bounding box in SOT and by a pixel-wise mask in VOS. Instead, in multiple object tracking settings (MOT [41], MOTS [57] and PoseTrack [2]), tracking means connecting sets of (often given) detections across video frames to address the problem of identity association and forming trajectories. Despite these tasks only differing in the number of objects per frame to consider and observation format (bounding boxes, keypoints or masks), the best practices developed by the methods tackling them vary significantly.

---

*Corresponding author

35th Conference on Neural Information Processing Systems (NeurIPS 2021).

Though the proliferation of setups, benchmarks and methods is positive in that it allows specific use cases to be thoroughly studied, we argue it makes increasingly harder to effectively study one of the fundamental problems that all these tasks have in common, *i.e. what constitutes a good representation to track objects throughout a video?* Recent advancements in large-scale models for language [15, 6] and vision [24, 10] have suggested that a strong representation can help addressing multiple downstream tasks. Similarly, we speculate that a good representation is likely to benefit many different tracking tasks, regardless of their specific setup. In order to validate our speculation, in this paper we present a framework that allows to adopt the same appearance model to address *five* different tracking tasks (Figure 2). In our taxonomy (Figure 4), we consider existing tracking tasks as problems that have either *propagation* or *association* at their core. When the core problem is propagation (as in SOT and VOS), one has to localise a target object in the current frame given its location in the previous one. Instead, in association problems (MOT, MOTS, and PoseTrack), target states in both previous and current frames are given, and the goal is to determine the correspondence between the two sets of observations. We show how most tracking tasks currently considered by the community can be simply expressed starting from the *primitives* of propagation or association. For propagation tasks, we employ existing box and mask propagation algorithms [5, 61, 58]. For association tasks, we propose a novel reconstruction-based metric that leverages fine-grained correspondence to measure similarities between observations. In the proposed framework, each individual task is assigned to a dedicated "head" that allows to represent the object(s) in the appropriate format to compare against prior arts on the relevant benchmarks.

Note that, in our framework, only the appearance model contains parameters that can be learned via back-propagation, and that we do not experiment with appearance models that have been trained on specific tracking tasks. Instead, we adopt models trained via recent self-supervised learning (SSL) techniques and that have already demonstrated their effectiveness on a variety of image-based tasks. Our motivation is twofold. First, SSL models are particularly interesting for our use-case, as they are explicitly conceived to be of general purpose. As a byproduct, our work also serves the purpose of evaluating and comparing appearance models obtained from self-supervised learning approaches (see Figure 1). Second, we hope to facilitate the tracking community in directly benefiting from the rapid advancements of the self-supervised learning literature.

To summarise, the contributions of our work are as follows:

- We propose UniTrack, a framework that supports five tracking tasks: SOT [70], VOS [48], MOT [41], MOTS [57], and PoseTrack [2]; and that can be easily extended to new ones.
- We show how UniTrack can leverage many existing general-purpose appearance models to achieve a performance that is competitive with the state-of-the-art on several tracking tasks.
- We propose a novel reconstruction-based similarity metric for association that preserves fine-grained visual features and supports multiple observation formats (box, mask and pose).
- We perform an extensive evaluation of self-supervised models, significantly extending the empirical analysis of prior literature to video-based tasks.

## 2 The UniTrack Framework

### 2.1 Overview

Inspecting existing tracking tasks and benchmarks, we noticed that their differences can be roughly categorised across four axes, illustrated in Figure 2 and detailed below.

1. Whether the requirement is to track a single object (SOT [70, 29], VOS [48]), or multiple objects (MOT [48], MOTS [57], PoseTrack [2]).
2. Whether the targets are specified by a user in the first frame only (SOT, VOS), or instead are given in every frame, *e.g.* by a pre-trained detector (MOT, MOTS, PoseTrack).
3. Whether the target objects are represented by bounding-boxes (SOT, MOT), pixel-wise masks (VOS, MOTS) or pose annotations (PoseTrack).
4. Whether the task is class-agnostic, *i.e.* the target objects can be of *any* class (SOT, VOS); or if instead they are from a predefined set of classes (MOT, MOTS, PoseTrack).

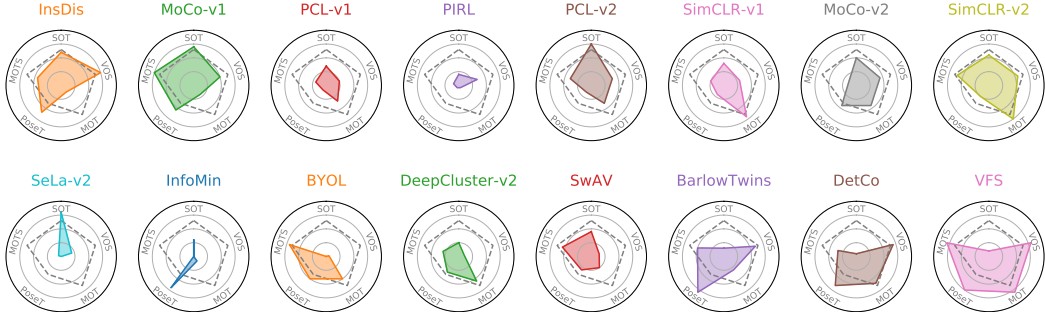

**Figure 1:** High-level overview of the performance of sixteen self-supervised learning models on five tracking tasks: SOT, VOS, MOT, PoseTracking and MOTS. A higher rank (better performance) corresponds to a vertex nearer to the outer circle. A larger area of the pentagon signifies better overall performance of its respective appearance model. Results of a vanilla ImageNet-supervised model are indicated with a gray dashed line as reference. Notice how the best model VFS [74] dominates on four out of the five tasks considered.

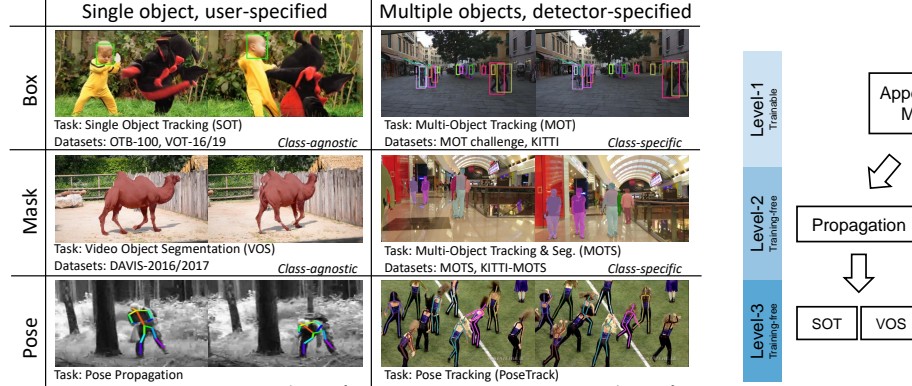

**Figure 2:** Existing tracking problems and their respective benchmarks differ from each other under several aspects: the assumption could be that there is a single or multiple objects to track; targets can be specified by the user in the first frame only, or assumed to be given at every frame (*e.g.* provided by a detector); the classes of the targets can be known (class-specific) or unknown (class-agnostic); the representation of the targets can be bounding boxes, pixel-wise masks, or pose annotations.

**Figure 3:** Overview of UniTrack. The framework can be divided in three levels. Level-1: a trainable appearance model. Level-2: the fundamental primitives of *propagation* and *association*. Level-3: task-specific heads.

Typically, in single-object tasks the target is specified by the user in the first frame, and it can be of any class. Instead, for multi-object tasks detections are generally considered as given for every frame, and the main challenge is to solve identity association for the several objects. Moreover, in multi-object tasks the set of classes to address is generally known (*e.g.* pedestrians or cars).

Figure 3 depicts a schematic overview of the proposed UniTrack framework, which can be understood as conceptually divided in three "levels". The first level is represented by the appearance model, responsible for extracting high-resolution feature maps from the input frame (Section 2.2). The second level consists of the algorithmic primitives addressing *propagation* (Section 2.3) and *association* (Section 2.4). Finally, the last level comprises multiple task-specific algorithms that make direct use of the primitives of the second level. In this work, we illustrate how UniTrack can be used to obtain competitive performance on all of the five tracking tasks of level-3 from Figure 3. Moreover, new tracking tasks can be easily integrated.

Importantly, note that the appearance model is the only component containing trainable parameters. The reason we opted for a shared and non task-specific representation is twofold. Firstly, the large amount of different setups motivated us to investigate whether having separately-trained models for each setup is necessary. Since training on specific datasets can bias the representation towards a limited set of visual concepts (*e.g.* animals or vehicles) and limit its applicability to "open-world" settings, we wanted to understand how far can a shared representation go. Second, we wanted to

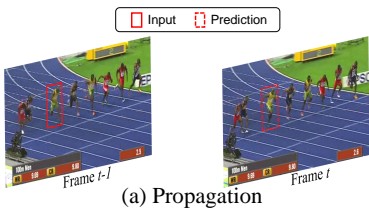
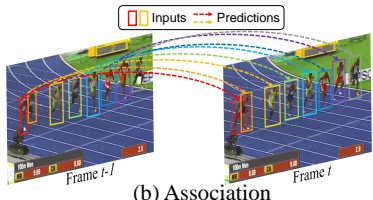

(a) Propagation            (b) Association

**Figure 4:** Propagation *v.s.* Association. In the *propagation* problem, the goal is to estimate the target state at the current frame given the observation in the previous one. This is typically addressed for one object at the time. In the *association* problem, observations in both previous and current frames are given, and the goal is to determine correspondences between the two sets.

provide the community with multiple baselines that can be used to better assess newly proposed contributions, and that can be immediately used on new datasets and tasks without the need of retraining.

## 2.2 *Base* appearance model

The base appearance model $\phi$ takes as input a 2D image $I$ and outputs a feature map $X = \phi(I) \in \mathbb{R}^{H \times W \times C}$. Since ideally an appearance model used for object propagation and association should be able to leverage fine-grained semantic correspondences between images, we choose a network with a small stride of $r = 8$, so that its output in feature space can have a relatively large resolution.

We refer to the vector (along the channel dimension) of a single point in the feature map as a *point vector*. We expect a point vector $x_1^i \in \mathbb{R}^C$ from the feature map $X_1$ to have a high similarity with its "true match" point vector $x_2^{\hat{i}}$ in $X_2$, while being far apart from all the other point vectors $x_2^j$ in $X_2$; *i.e.* we expect $s(x_1^i, x_2^{\hat{i}}) > s(x_1^i, x_2^j), \forall j \neq \hat{i}$, where $s(\cdot, \cdot)$ represents a similarity function.

In order to learn fine-grained correspondences, fully-supervised methods are only amenable for synthetic datasets (*e.g.* Flying Chairs for optical flow [16]). With real-world data, it is intractable to label pixel-level correspondences and train models in a fully-supervised fashion. To overcome this obstacle, in this paper we adopt representations obtained with self-supervision. We experiment both with models trained with approaches that leverage pixel-wise pretext tasks [27, 58] and, inspired by prior works that have pointed out how fine-grained correspondences emerge in middle-level features [39, 74], with models obtained from image-level tasks (*e.g.* MoCo [24], SimCLR [10]).

## 2.3 Propagation

**Problem definition.** Figure 4a schematically illustrates the problem of *propagation*, which we use as a primitive to address SOT and VOS tasks. Considering the single-object case, given video frames $\{I_t\}_{t=1}^T$ and an initial ground truth observation $z_1$ as input, the goal is to predict object states $\{\hat{z}_t\}_{t=2}^T$ for each time-step $t$. In this work we consider three formats to represent objects: bounding boxes, segmentation masks and pose skeletons.

**Mask propagation.** In order to propagate masks, we rely on the approach popularised by recent video self-supervised methods [27, 58, 35, 31]. Consider the feature maps of a pair of consecutive frames $X_{t-1}$ and $X_t$, both $\in \mathbb{R}^{s \times C}$, and the label mask $z_{t-1} \in [0, 1]^s$ of the previous frame [2], where $s = H \times W$ indicates its spatial resolution. We compute the matrix of transitions $K_{t-1}^t = [k_{i,j}]_{s \times s}$ as the affinity matrix between $X_{t-1}$ and $X_t$. Each element $k_{i,j}$ is defined as

$$k_{i,j} = \text{Softmax}(X_{t-1}, X_t^\top; \tau)_{ij} = \frac{\exp(\langle x_{t-1}^i, x_t^j \rangle / \tau)}{\sum_k^s \exp(\langle x_{t-1}^i, x_t^k \rangle / \tau)}, \tag{1}$$

where $\langle \cdot, \cdot \rangle$ indicates inner product, and $\tau$ is a temperature hyperparameter. As in [27], we only keep the top $K$ values for each row and set other values to zero. Then, the mask for the current frame at time $t$ is predicted by propagating the previous prediction: $z_t = K_{t-1}^t z_{t-1}$. Mask propagation proceeds in a recurrent fashion: the output mask of the current frame is used as input for the next one.

---

[2]Note this corresponds to the ground-truth initialisation when $t = 1$, and to the latest prediction otherwise.

**Pose propagation.** In order to represent pose keypoints, we use the widely adopted Gaussian *belief maps* [66]. For a keypoint $p$, we obtain a belief map $z^p \in [0,1]^s$ by using a Gaussian with mean equal to the keypoint's location and variance proportional to the subject's body size. In order to propagate a pose, we can then individually propagate each belief map in the same manner as mask propagation, again as $z_t^p = K_{t-1}^t z_{t-1}^p$.

**Box propagation.** The position of an object can also be more simply expressed with a four-dimensional vector $z = (u, v, w, h)$, where $(u, v)$ are the coordinates of the bounding-box center, and $(w, h)$ are its width and height. While one could reuse the strategy adopted above by simply converting the bounding-box to a pixel-wise mask, we observed that using this strategy leads to inaccurate predictions. Instead, we use the approach of SiamFC [5], which consists in performing cross-correlation (XCORR) between the target template $z_{t-1}$ and the frame $X_t$ to find the new location of the target in frame $t$. Cross-correlation is performed at different scales, so that the bounding-box representation can be resized accordingly. We also provide a Correlation Filter-based alternative (DCF) [54, 61] (see Appendix B.1).

## 2.4 Association

**Problem definition.** Figure 4b schematically illustrates the *association* problem, which we use as primitive to address the tasks of MOT, MOTS and PoseTrack. In this case, observations for object states $\{\hat{\mathcal{Z}}_t\}_{t=1}^T$ are given for all the frames $\{I_t\}_{t=1}^T$, typically via the output of a pre-trained detector. The goal here is to form trajectories by connecting observations across adjacent frames according to their identity.

**Association algorithm.** We adopt the association algorithm proposed in JDE [65] for MOT, MOTS and PoseTrack tasks, of which detailed description can be found in Appendix C.1. In summary, we compute an $N \times M$ distance matrix between $N$ already-existing tracklets and $M$ "new" detections from the last processed frame. We then use the Hungarian algorithm [30] to determine pairs of matches between tracklets and detections, using the distance matrix as input. To obtain the matrix of distances used by the algorithm, we compute the linear combination of two terms accounting for *motion* and *appearance* cues. For the former, we compute a matrix indicating how likely a detection corresponds to the object state predicted by a Kalman Filter [28]. Instead, the appearance component is directly computed by using feature-map representations obtained by processing individual frames with the appearance model (Section 2.2). While object-level features for box and mask observations can be directly obtained by cropping frame-level feature maps, when an object is represented via a pose it first needs to be converted to a mask (via a procedure described in Appendix C.2).

A key issue of this scenario is how to measure similarities between object-level features. We find existing methods limited. First, objects are often compared by computing the cosine similarity of average-pooled object-level feature maps [84, 51]. However, the operation of average inherently discards local information, which is important for fine-grained recognition. Approaches [18, 52] that instead to some extent do preserve fine-grained information, such as those computing the cosine similarity of (flattened) feature maps, do not support objects with differently-sized representation (situation that occurs for instance with pixel-level masks). To cope with the above limitations, we propose a reconstruction-based similarity metric that is able to deal with different observation formats, while still preserving fine-grained information.

**Reconstruction Similarity Metric (RSM).** Let $\{t_i\}_{i=1}^N$ denote the object-level features of $N$ existing tracklets, $t_i \in \mathbb{R}^{s_{t_i} \times C}$ and $s_{t_i}$ indicates the spatial size of the object, *i.e.* the area of the box or the mask representing it. Similarly, $\{d_j\}_{j=1}^M$ denotes the object-level features of $M$ new detections. With the goal of computing similarities to obtain an $N \times M$ affinity matrix to feed to the Hungarian algorithm, we propose a novel reconstruction-based similarity metric (RSM) between pairs $(i, j)$, which is obtained as

$$\text{RSM}(i, j) = \frac{1}{2}(\cos(t_i, \hat{t}_{i \leftarrow j}) + \cos(d_j, \hat{d}_{j \leftarrow i})), \tag{2}$$

where $\hat{t}_{i \leftarrow j}$ represents $t_i$ reconstructed from $d_j$ and $\hat{d}_{j \leftarrow i}$ represents $d_j$ reconstructed from $t_i$. In multi-object tracking scenarios, observations are often incomplete due to frequent occlusions. As such, directly comparing features between incomplete and complete observations often fails because of misalignment between local features. Suppose $d_j$ is a detection feature representing a severely occluded pedestrian, while $t_i$ a tracklet feature representing the same person, but unoccluded. Likely,

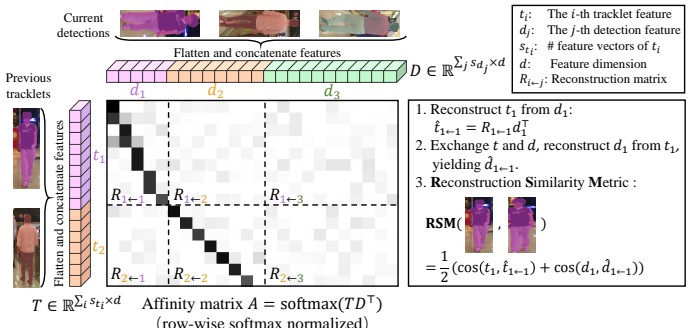

**Figure 5:** *Reconstruction Similarity Metric* (RSM): First, object-level features of existing tracklets and current detections are flattened and concatenated. Then, an affinity matrix between the two feature sets is computed. For a pair of tracklet $t_i$ and detection $d_j$, we "extract" the corresponding sub-matrix from the entire affinity matrix as linear weights and reconstruct $t_i$ from $d_j$ using these linear weights. The similarity between the original object-level feature and its reconstructed version is finally taken as the RSM. We want the metric to be symmetric, so we perform reconstruction both forward ($t_i \leftarrow d_j$) and backward ($t_i \rightarrow d_j$).

directly computing the cosine similarity between the two will not be very telling. RSM addresses this issue by introducing a step of reconstruction after which the co-occurring parts of point features will be better aligned, thus making the final similarity more likely to be meaningful.

The reconstructed object-level feature map $\hat{t}_{i \leftarrow j}$ is a simple linear transformation of $d_j$, *i.e.* $\hat{t}_{i \leftarrow j} = R_{i \leftarrow j} d_j$, where $R_{i \leftarrow j} \in \mathbb{R}^{s_{t_i} \times s_{d_j}}$ is a transformation matrix obtained as follows. We first flatten and concatenate all object-level features belonging to a tracklet (*i.e.* the set of observations corresponding to an object) into a single feature matrix $T \in \mathbb{R}^{(\sum_i s_{t_i}) \times C}$. Similarly, we obtain all the object-level feature maps of a new set of detections $D \in \mathbb{R}^{(\sum_j s_{d_j}) \times C}$. Then, we compute the affinity matrix $A = \text{Softmax}(TD^\top)$ and "extract" individual $R_{i \leftarrow j}$ mappings as sub-matrices of $A$ with respect to the appropriate $(i, j)$ tracklet-detection pair: $R_{i \leftarrow j} = A \left[ \sum_{i'=1}^{i-1} s_{i'} : \sum_{i'=1}^{i} s_{i'}, \sum_{j'=1}^{j-1} s_{j'} : \sum_{j'=1}^{j} s_{j'} \right]^3$. For a schematic representation of the procedure just described, see Figure 5.

RSM can be interpreted from an *attention* [55] perspective. The feature map of a tracklet $t_i$ being reconstructed can be seen as a set of *queries*, and the "source" detection feature $d_j$ can be interpreted both as *keys* and *values*. The goal is to reconstruct the queries by linear combination of the values. The linear combination (attention) weights are computed using the affinity between queries and keys. Specifically, we first compute a *global* affinity matrix between $t_i$ and all the $d_{j'}$ for $j' = 1, ..., M$, and then extract the corresponding sub-matrix for $t_i$ and $d_{j'}$ as the attention weights. Our formulation leads to a desired property: if the attention weights approach zero, the corresponding reconstructed point vectors will approach zero and so the RSM between $t_i$ and $d_j$.

Measuring similarity by reconstruction is popular in problems such as few-shot learning [67, 80], self-supervised learning [38], and person re-identification [26]. However, reconstruction is typically framed as a ridge regression or optimal transport problem. With $O(n^2)$ complexity, RSM is more efficient than ridge regression and it has a similar computation cost to calculating the Earth Moving Distance for the optimal transport problem. Appendix D shows a series of ablation studies illustrating the importance of the proposed RSM for the effectiveness of UniTrack on association-type tasks.

## 3 Experiments

Since UniTrack does not require task-specific training, we were able to experiment with many alternative appearance models (see Figure 3) with little computational cost. In Section 3.1 we perform an extensive evaluation to benchmark a wide variety of off-the-shelf, modern self-supervised models, showing their strengths and weaknesses on all five tasks considered. In this section we also conduct a correlation study with the so-called "linear probe" strategy [81], which became a popular

---

[3]Here we use a numpy-style matrix slicing notation to represent a submatrix, *i.e.* $A[i : j, k : l]$ indicates a submatrix of $A$ with row indices ranging from $i$ to $j$ and column indices ranging from $k$ to $l$.

way to evaluate representations obtained with self-supervised learning. Then, in Section 3.2 we compare UniTrack (equipped with supervised or unsupervised appearance models) against recent and task-specific tracking methods.

**Implementation details.** We use ResNet-18 [25] or ResNet-50 as the default architecture. With ImageNet-supervised appearance model, we refer to the ImageNet pre-trained weights made available in PyTorch's "Model Zoo". To prevent excessive downsampling, we modify the spatial stride of `layer3` and `layer4` to 1, achieving a total stride of $r = 8$. We extract features from both `layer3` and `layer4`. We report results with `layer3` features when comparing against task-specific methods (Section 3.2), and with both `layer3` and `layer4` when evaluating multiple different representations (Section 3.1). Further implementation details are deferred to Appendix B and C.

**Datasets and evaluation metrics.** For fair comparison with existing methods, we report results on standard benchmarks with conventional metrics for each task. Please refer to Appendix A for details.

### 3.1 UniTrack as evaluation platform of previously-learned representations

The process of evaluating representations obtained via self-supervised learning (SSL) often involves additional training [17, 24, 10], for instance via the use of *linear probes* [81], which require to fix the pre-trained model and train an additional linear classifier on top of it. In contrast, using UniTrack as evaluation platform **(1)** does not require any additional training and **(2)** enables the evaluation on a battery of important video tasks, which have generally been neglected in self-supervised-learning papers in favour of more established image-level tasks such as classification.

In this section, we evaluate three types of SSL representations: **(a)** Image-level representations learned from images, *e.g.* MoCo [24] and BYOL [21]; **(b)** Pixel-level representations learned from images (such as DetCo [72] and PixPro [73]) and **(c)** videos (such as UVC [35] and CRW [27]). For all methods considered, we use the pre-trained weights provided by the authors.

Results are shown in Table 1 and 2, where we report the results obtained by using features from either `layer3` or `layer4` of the pre-trained ResNet backbone. We report both results and separate them by a '/' in the table. Note that, for this analysis only, for association-type tasks motion cues are discarded to better highlight distinctions between different representations and avoid potential confounding factors. Figure 1 provides a high-level summary of the results by focusing on the ranking obtained by different SSL methods on the five tasks considered (each represented by a vertex in the radar-style plot). Several observations can be made:

**(1)** *There is no significant correlation between "linear probe accuracy" on ImageNet and overall tracking performance.* The linear probe approach [81] has become a standard way to compare SSL representations. In Figure 6, we plot tracking performance on five tasks (y-axes) against ImageNet top-1 accuracy of 16 different models (x-axes), and report Pearson and Spearman (rank) correlation coefficients. We observe that the correlation between ImageNet accuracy and tracking performance is small, *i.e.* the Pearson's $r$ ranges from $-0.38$ to $+0.20$, and Spearman's $\rho$ ranges from $-0.36$ to $+0.26$. For most tasks, there is almost no correlation, while for VOS the two measures are mildly *inversely* correlated. The result suggests that evaluating SSL models on five extra tasks with UniTrack could constitute a useful complement to ImageNet linear probe evaluation, and encourage the SSL community to pursue the design of even more *general purpose* representations.

**(2)** *A vanilla ImageNet-trained supervised representation is surprisingly effective across the board.* On most tasks, it reports a performance competitive with the best representation for that task. This is particularly evident from Figure 1, where its performance is outlined as a gray dashed line. This result suggests that results obtained with vanilla ImageNet features should be reported when investigating new tracking methods.

**(3)** *The best self-supervised representation ranks first on most tasks.* Recently, it has been shown how SSL-trained representations can match or surpass their supervised counterparts on ImageNet classification (*e.g.* [21]) and many downstream tasks [17, 72]. Within UniTrack, although no individual SSL representation is able to beat the vanilla ImageNet-trained representation on every single task, we observe that the recently proposed VFS [74] ranks first on every task, except for single-object tracking. This suggests that advancements of the self-supervised learning literature can directly benefit the tracking community: it is reasonable to expect that newly-proposed representations will further improve performance across the board.

| Representation | SOT [70] | | VOS [48] | MOT [41] | | MOTS [57] | | PoseTrack [2] | |
|---|---|---|---|---|---|---|---|---|---|
| | $AUC_{XCorr}$ ↑ | $AUC_{DCF}$ ↑ | $\mathcal{J}$-mean↑ | IDF1↑ | HOTA↑ | IDF1↑ | HOTA↑ | IDF1↑ | IDs↓ |
| *Random Init.* | 10.3 / 9.0 | 28.0 / 20.0 | 29.3 / **33.9** | 8.4 / **8.9** | 8.4 / **8.5** | 20.8 / **23.1** | 25.9 / **28.7** | 40.2 / 38.5 | 88792 / 90963 |
| *ImageNet-sup.* | **58.6** / 49.5 | **62.0** / 53.9 | **62.3** / 57.9 | **75.6** / 73.2 | **63.3** / 61.8 | 68.4 / **69.4** | 70.2 / **71.0** | **73.7** / 73.3 | **6969** / 7103 |
| InsDis [71] | 47.6 / 47.3 | **61.8** / 51.1 | **62.6** / 60.1 | 66.7 / **73.9** | 57.9 / **61.9** | **68.4** / 68.0 | 69.6 / **70.3** | 72.4 / **73.9** | 7106 / **7015** |
| MoCoV1 [24] | 50.9 / 47.9 | **62.2** / 53.7 | **61.5** / 57.9 | 69.2 / **74.1** | 59.4 / **61.9** | **70.6** / 69.3 | **71.6** / 70.9 | 72.8 / **73.9** | **6872** / 7092 |
| PCLV1 [34] | **56.8** / 31.5 | **61.3** / 35.0 | **60.4** / 38.8 | **74.8** / 68.8 | **62.8** / 59.1 | **67.6** / 65.2 | **69.7** / 67.3 | **73.3** / 71.1 | **6855** / 10694 |
| PIRL [42] | 43.8 / **51.0** | **61.2** / 53.4 | **60.8** / 57.7 | 62.0 / **73.4** | 54.6 / **61.9** | 66.0 / **67.4** | 66.7 / **69.9** | 72.1 / **73.0** | 7235 / **7173** |
| PCLV2 [34] | **54.9** / 50.3 | **62.5** / 51.6 | **61.2** / 52.5 | **74.9** / 72.9 | **62.7** / 61.8 | **68.3** / 66.6 | **70.5** / 69.0 | **73.5** / 73.4 | **6859** / 8489 |
| SimCLRV1 [10] | 47.3 / **51.9** | **61.3** / 50.7 | **60.5** / 56.5 | 66.9 / **75.6** | 57.7 / **63.2** | 65.8 / **67.6** | 67.7 / **69.5** | 72.3 / **73.5** | 7084 / 7367 |
| MoCoV2 [12] | **53.7** / 47.2 | **61.5** / 53.3 | **61.2** / 54.0 | 72.0 / **74.9** | 61.2 / **62.8** | **67.5** / 67.3 | **69.6** / 69.6 | 73.0 / **73.7** | **6932** / 7702 |
| SimCLRV2 [11] | 50.0 / **54.7** | **61.7** / 56.8 | **61.6** / 58.4 | 67.6 / **75.7** | 58.1 / **63.3** | **69.1** / 67.4 | **70.4** / 69.4 | 72.5 / **73.6** | 7228 / 7856 |
| SeLaV2 [3] | **51.0** / 9.6 | **63.1** / 14.2 | **60.2** / 40.2 | 68.8 / **68.9** | 59.0 / **59.3** | **66.8** / 66.1 | **68.7** / 68.5 | **72.9** / 72.3 | **6983**/ 7815 |
| Infomin [53] | **48.5** / 46.8 | **61.2** / 51.9 | **58.4** / 51.1 | 66.7 / **73.4** | 57.6 / **61.9** | **66.7** / 66.3 | 68.5 / **68.8** | 72.5 / **74.0** | 7066 / 7901 |
| BarLow [79] | 44.5 / **55.5** | **60.5** / 60.1 | **61.7** / 57.8 | 63.7 / **74.5** | 55.4 / **62.4** | **68.7** / 67.4 | 69.5 / **69.8** | 72.3 / **74.3** | 7131 / 7456 |
| BYOL [21] | 48.3 / **55.5** | **58.9** / 56.8 | **58.8** / 54.3 | 65.3 / **74.9** | 56.8 / **62.9** | **70.1** / 66.8 | **70.8** / 69.3 | 72.4 / **73.8** | 7213 / 8032 |
| DeepCluster [7] | 51.5 / **52.9** | 61.2 / **61.2** | **59.3** / 53.4 | 66.9 / **75.1** | 57.8 / **63.5** | **67.7** / 67.4 | 69.4 / **69.8** | 72.7 / **73.7** | 7018 / 7283 |
| SwAV [8] | 49.2 / **52.4** | **61.5** / 59.4 | **59.4** / 57.0 | 65.6 / **74.4** | 56.9 / **62.3** | **68.8** / 67.0 | 69.9 /69.5 | 72.7 / **73.6** | 7025 / 7377 |
| VFS [74] | **51.1** / 45.3 | **60.3** / 43.8 | **62.8** / 56.8 | 74.1 / **77.0** | 62.6 / **63.9** | **71.0** / 68.0 | **72.1** / 70.4 | 73.3 / **74.2** | **6731** / 7091 |
| PixPro [73] | 40.5 / **49.2** | **57.4** / 49.3 | **56.4** / 52.2 | 61.7 / **67.7** | 54.3 / **58.6** | 64.2 / **66.2** | 65.1 / **67.6** | 72.4 / **73.1** | 7163 / **6953** |
| DetCo [72] | **55.0** / 47.1 | **59.0** / 53.2 | **62.3** / 56.1 | **75.3** / 72.9 | **62.8** / 61.6 | **67.8** / 66.8 | **70.0** / 69.4 | **73.9** / 73.3 | 7357 / 8009 |
| TimeCycle [64] | **43.8** / 24.2 | **57.5** / 48.7 | **51.8** / 48.9 | **68.7** / 28.2 | **59.3** / 25.5 | **69.9** / 47.1 | **71.3** / 49.3 | **72.0** / 62.3 | 7837 / 27884 |

**Table 1:** Tracking performance of pre-trained *image-based* SSL models. All methods employ a ResNet-50.

| Representation | SOT [70] | | VOS [48] | MOT [41] | | MOTS [57] | | PoseTrack [2] | |
|---|---|---|---|---|---|---|---|---|---|
| | $AUC_{XCorr}$ ↑ | $AUC_{DCF}$ ↑ | $\mathcal{J}$-mean↑ | IDF1↑ | HOTA↑ | IDF1↑ | HOTA↑ | IDF1↑ | IDs↓ |
| *Random Init.* | 16.0 / **18.2** | **36.1** / 32.1 | 33.0 / **36.7** | **18.4** / 14.6 | **20.2** / 12.9 | **34.5** / 33.1 | **39.9** / 37.6 | **52.8** / 50.5 | 65317 / 66230 |
| *ImageNet-sup.* | **55.0** / 46.2 | **61.8** / 52.6 | **58.4** / 46.7 | **74.8** / 74.5 | **62.7** / 62.1 | 67.6 / **68.6** | 69.8 / **70.5** | 72.7 / **73.2** | 6808 / 7024 |
| Color. [58]+mem. | 41.6 / **43.4** | 56.7 / **58.7** | 53.6 / **59.7** | **64.9** / 62.8 | **56.8** / 55.5 | **68.8** / 66.1 | **69.4** / 66.3 | 72.4 / **72.6** | 6850 / **6778** |
| UVC [35] | **46.0** / 38.7 | 58.1 / **59.9** | **56.5** / 53.9 | **66.9** / 64.5 | **57.7** / 54.1 | **69.9** / 68.7 | 69.6 / 69.4 | 72.6 / **72.8** | 6843 / 6972 |
| CRW [27] | 46.3 / **49.1** | **58.9** / 54.9 | **63.2** / 60.7 | 67.8 / **73.0** | 58.4 / **61.7** | 69.0 / **71.3** | 69.2 / **71.9** | 72.7 / **73.0** | 6799 / **6761** |

**Table 2:** Tracking performance of pre-trained *video-based* SSL models. All methods employ a ResNet-18. In the above two tables, we report results with [layer3 / layer4] features in each cell, and the best performance between the two is **bolded**. We use the bolded values to rank the models in each column, and visualise (column-wise) better performance with darker cell colors. Best results in each column are underlined.

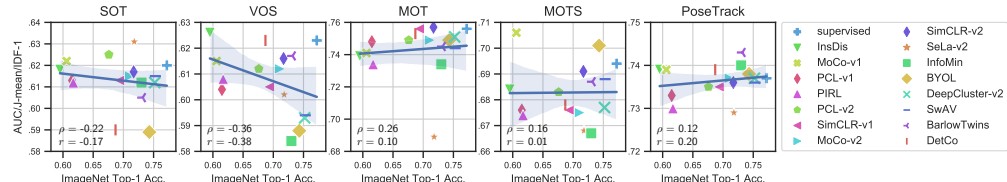

**Figure 6:** Tracking performance is poorly correlated with ImageNet accuracy. On the x-axes we plot ImageNet linear probe top-1 accuracy and on the y-axes the tracking performance on five tracking datasets. Correlation coefficients (Spearman's $\rho$ and Pearson's $r$) are shown in the left bottom of each plot.

**(4)** *Pixel-level SSL representations do not seem to have a consistent advantage in pixel-level tasks.* In Table 2 and at the bottom of Table 1 we compare recent SSL representations trained with pixel-level proxy tasks: PixPro [73], DetCo [72], TimeCycle [64], Colorization [58], UVC [35] and Contrastive Random Walk (CRW) [27]. Considering that pixel-level models leverage more fine-grained information during training, one may expect them to outperform image-based models in the tracking tasks where this is important. It is not straightforward to compare pixel-level SSL models with image-level ones, as the two types employ different default backbone networks. However, note how good image-based models (MoCo-v1, SimCLR-v2) are on par with their supervised counterpart in all tasks, while good pixel-level models (DetCo, CRW) still have gaps with respect to their supervised counterparts in tasks like SOT and MOT. Moreover, from Table 1, one can notice how the last three rows, despite representing methods leveraging pixel-level information during training, are actually outperformed by image-level representations on the pixel-level tasks of VOS, MOTS and PoseTrack.

**(5)** *Video data can benefit representation learning for video tasks.* The top-ranking VFS is similar to MoCo, SimCLR and BYOL in terms of learning scheme: they all perform contrastive learning on image level features. The most important distinction is the training data. Previous SSL methods mostly train on still-image based datasets (typically ImageNet), while VFS employs a large-scale video dataset Kinetics [9]. Clearly, this is not very surprising, as training on video data can help closing the domain gap with the (video-based) downstream tasks considered in this paper.

| Methods | IDF1↑ | IDs↓ | MOTA↑ | HOTA↑ |
|---|---|---|---|---|
| JDE [65] | 55.8 | 1544 | 64.4 | - |
| CTracker [47] | 57.2 | 1897 | 67.6 | 48.8 |
| TubeTK [46] | 62.2 | 1236 | 66.9 | 50.8 |
| MAT [23] | 63.8 | 928 | 73.5 | 56.3 |
| TraDes [69] | 64.7 | 1144 | 70.1 | 53.2 |
| CSTrack [36] | 71.8 | 1071 | 70.7 | 59.8 |
| FairMOT† [82] | **72.8** | 1074 | **74.9** | **61.6** |
| UniTrack_ImageNet† | 71.8 | **683** | 74.7 | 59.1 |
| UniTrack_VFS† | 70.3 | 829 | 72.7 | 58.6 |

**(a)** MOT@MOT-16 [41] *test* split, private detection.

| Methods | IDF1↑ | IDs↓ | sMOTA↑ |
|---|---|---|---|
| TrackRCNN [57] | 42.4 | 567 | 40.6 |
| SORTS [68] | 57.3 | 577 | 55.0 |
| PointTrack [76] | 42.9 | 868 | 62.3 |
| GMPHD [50] | 65.6 | 566 | 69.0 |
| COSTA† [1] | **70.3** | 421 | 69.5 |
| UniTrack_ImageNet† | 67.2 | 622 | 68.9 |
| UniTrack_VFS† | 68.2 | **342** | 69.7 |

**(b)** MOTS@MOTS [57] *test* split.

| Methods | IDF1↑ | IDs↓ | MOTA↑ |
|---|---|---|---|
| MDPN [22] | - | - | 50.6 |
| OpenSVAI [44] | - | - | 62.4 |
| Miracle [78] | - | - | 64.0 |
| KeyTrack [49] | - | - | **66.6** |
| TWVA [19] | - | - | 64.7 |
| LightTrack† [43] | 52.2 | **3024** | 64.8 |
| UniTrack_ImageNet† | 73.2 | 6760 | 63.5 |
| UniTrack_VFS† | **74.2** | 7091 | 63.3 |

**(c)** PoseTrack@PoseTrack2018 [2] *val* split.

| Methods | $\mathcal{J}$-mean↑ |
|---|---|
| *Supervised:* | |
| SiamMask [62] | 54.3 |
| FEELVOS [56] | 63.7 |
| STM [45] | **79.2** |
| *Unsupervised:* | |
| Colorization [58] | 34.6 |
| TimeCylce [64] | 40.1 |
| UVC [35] | 56.7 |
| CRW [27] | 64.8 |
| UniTrack_ImageNet | 58.4 |
| UniTrack_VFS | 62.8 |

**(d)** VOS@DAVIS-2017 [48].

| Methods | AUC↑ |
|---|---|
| *Supervised:* | |
| SiamFC [5] | 58.2 |
| SiamRPN [33] | 63.7 |
| SiamRPN++ [32] | **69.6** |
| *Unsupervised:* | |
| UDT [59] | 59.4 |
| UDT+ [59] | 63.2 |
| LUDT [60] | 60.2 |
| LUDT+ [60] | **63.9** |
| UniTrack_ImageNet_XCorr | 55.5 |
| UniTrack_ImageNet_DCF | 61.8 |
| UniTrack_VFS_DCF | 60.3 |

**(e)** SOT@OTB-2015 [70].

**Table 3:** Comparison with task-tailored unsupervised and supervised methods on five typical tracking tasks. † indicates methods using identical observations.

### 3.2 Comparison with task-specific tracking methods

**Unsupervised methods.** We observe that UniTrack performs competitively against unsupervised state-of-the-art methods in both the propagation-type tasks we considered (Table 3d and 3e). For SOT, UniTrack with a DCF head [61] outperforms UDT [59] (a strong recent method) by 2.4 AUC points, while it is surpassed by LUDT+ [60] by 2.1 points. Considering that LUDT+ adopts an additional online template update mechanism [13] while ours does not, we believe the gap could be closed. In VOS, existing unsupervised methods are usually trained on video datasets [35, 27], and some of the most recent outperform UniTrack (with an ImageNet-trained representation). Nonetheless, when we use a VFS-trained representation, this performance difference is reduced to 2%. Finally, note that for association-type tasks we are not aware of any existing unsupervised learning method, and thus in this case we limit the comparison to supervised methods.

**Comparison with supervised methods.** In general, UniTrack with a ResNet-18 appearance model already performs on par with several existing task-specific supervised methods, and in several tasks it even shows superior accuracy, especially for identity-related metrics. **(1)** For SOT, UniTrack with a DCF head outperforms SiamFC [5] by 3.6 AUC points. This is a significant margin considering that SiamFC is trained with a large amount of crops from video datasets with annotated bounding boxes. **(2)** For VOS, UniTrack surpasses SiamMask [62] by 4.1 $\mathcal{J}$-mean points, despite this being trained on the joint set of three large-scale video datasets [37, 14, 75]. **(3)** For MOT, we employ the same detections used by the state-of-the-art tracker FairMOT [82]. The appearance embedding in FairMOT is trained with 270K bounding boxes of 8.7K labeled identities, from a MOT-specific dataset. In contrast, despite our appearance model not being trained with any MOT-specific data, our IDF1 score is quite competitive (71.8 *v.s.* 72.8 of FairMOT), and the ID switches are considerably reduced by 36.4%, from 1074 to 683. **(4)** For MOTS, we start from the same segmentation masks used by the COSTA [1] tracker, and observe a degradation in terms of ID switches (622 vs the 421 of the state of the art), and also a gap in IDF1 and sMOTA. **(5)** Finally, for pose tracking, we employ the same pose estimator used by LightTrack [43]. Compared with LightTrack, the MOTA of UniTrack degrades of 1.3 points because of an increased amount of ID switches. However, the IDF-1 score is improved by a significant margin (+21.0 points). This shows UniTrack preserves identity more accurately for long tracklets: even if ID switches occur more frequently, after a short period UniTrack is able to correct the wrong association, leading to a higher IDF-1.

Notice how, overall, UniTrack obtains more competitive performance on tasks that have association at their core, *i.e.* MOT, MOTS and PoseTrack. Upon inspection, we observed that most failure cases

in propagation-type tasks regard the "drift" occurring when the scale of the object is improperly estimated. In future work, this could be addressed for instance by a bounding-box regression module to refine predictions, or by carefully designing a motion model. For association-type tasks, the consequences of any type of inaccuracy are isolated to individual pairs of frames, and thus much less catastrophic by nature.

## 4 Related Work

To the best of our knowledge, **sharing the appearance model across multiple tracking tasks** has not been extensively studied in the computer vision literature, and especially not in the context of SSL representations. Some existing methods do share a common backbone architecture across tasks. For instance, STEm-Seg [4] addresses VIS [77] and MOTS; while TraDeS [69] addresses MOT, MOTS and VIS. However, both methods need to be trained separately and on different datasets for every task. Conversely, we reuse the same representation across five tasks. A promising direction for future work would be to use UniTrack to train a shared representation in a multi-task fashion. Only a few relevant works do adopt a multi-task approach [62, 83, 40], and they usually consider SOT and VOS tasks only. In general, despite the multi-task direction being surely interesting, it requires the availability of large-scale datasets with annotations in multiple formats, and costly training. These are two of the main reasons for which we believe that having a framework that allows to achieve competitive performance on multiple tasks with previously-trained models is a worthwhile endeavour.

**Self-supervised model evaluation**. Given the difference between the pretext tasks used to train self-supervised models and the downstream tasks used to evaluate them, the comparison between self-supervised approaches has always been a delicate matter. Existing evaluation strategies typically require additional training once a general-purpose representation has been obtained. One strategy keeps the representation fixed, and then trains additional task-specific heads with very limited capacity (*e.g.* a linear classifier [20, 10, 24] or a regression head for object detection [20]). A second strategy, instead, leverages SSL to obtain particularly effective initializations, and then proceeds to fine-tune such initialized models on the downstream task of interest. A wider range of tasks can be tested using this setup, such as semantic segmentation [17, 20] and surface normal estimation [20, 63]. In contrast, UniTrack provides a simpler way to evaluate SSL models, one that does not require additional training or fine-tuning. Also, this work is the first to extend SSL evaluation to a set of diverse video tasks. We believe this contribution will allow the study of self-supervised learning methods with a broader scope of applicability. Our work is also related to a line of self-supervised learning methods [27, 58, 35, 31] that learn their representations in a task-agnostic fashion, and then test it on *propagation* tasks (SOT and VOS). The design of UniTrack is inspired by their task-agnostic philosophy, while significantly extending their scope to a new set of tasks.

## 5 Conclusion

Do different tracking tasks require different appearance models? In order to address this question, the proposed UniTrack framework has been instrumental, as it has allowed to easily experiment with alternative representations on a wide variety of downstream problems. Although the answer is not a resounding "no", as only sometimes a single shared appearance model can outperform dedicated methods, we argue that a unified framework is an appealing alternative to task-specific methods. The main reason is that it allows us to make the most of the progress made in the representation learning literature at no extra cost. With the rapid development of self-supervised learning, and the large amount of computational resources dedicated to it, we believe it is reasonable to expect that, in the future, a general-purpose representation will be able to outperform task-specific methods across the board. Until then, UniTrack could still serve as a useful evaluation tool for novel representations, especially considering the lack of correlation with the standard linear-probe approach. We believe this will encourage the community to develop self-supervised representations that are of "general purpose" in a broader sense.

**Broader impact.** Upon reflection, we believe that progress in tracking applications and self-supervised learning is beneficial for society, as it can significantly impact (for instance) the development of autonomous vehicles, which we consider a net positive for society. We also recognise that the same technologies could constitute a threat if deployed for surveillance by entities hostile to civil liberties.

# 6 Funding Transparency Statement

This work was supported by the National Natural Science Foundation of China under Grant No. 61771288, Cross-Media Intelligent Technology Project of Beijing National Research Center for Information Science and Technology (BNRist) under Grant No. BNR2019TD01022 and the research fund under Grant No. 2019GQG0001 from the Institute for Guo Qiang, Tsinghua University.

This work was also supported by the EPSRC grant: Turing AI Fellowship: EP/W002981/1, EPSRC/MURI grant EP/N019474/1. We would also like to thank the Royal Academy of Engineering and FiveAI.

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
