# SUPPLEMENTARY MATERIAL
# Do Different Tracking Tasks Require Different Appearance Models?

## A  Datasets and Evaluation Metrics

The table below summarizes the datasets (all publicly available) and evaluation metrics used in this work. In general, to compare with existing task-specific methods, we use the most popular benchmark for each task and report the standard metrics.

For association-type tasks (MOT, MOTS and PoseTrack), we first report the MOTA metric since it highly-correlates with human's perception in measuring tracking accuracy [3]. However, the MOTA metric disproportionately overweights good detection accuracy [28, 8]. Since most multi-object trackers (included UniTrack) adopt off-the-shelf detectors, it is desirable to also adopt detection-independent measures of performance. For this reason, we also report identity based metrics such as IDF-1 and ID-switch. We also adopt the recently-introduced higher-order HOTA [28], to replace MOTA and to represent the overall tracking accuracy when comparing self-supervised methods.

For pose tracking, results are averaged for IDF-1 and MOTA, and summed for ID-switch, over 15 key points. In the main text, we only report results for the first five tasks from the table below. For the rest tasks (PoseProp and VIS) we provide additional results in Appendix E. We also provide SOT results on many more recent large-scale datasets in Appendix F.

| Task | SOT | VOS | MOT | MOTS | PoseTrack | PoseProp | VIS |
|------|-----|-----|-----|------|-----------|----------|-----|
| Dataset | OTB [50] | DAVIS 2017 [32] | MOT 16 [29] | MOTS [43] | PoseTrack 2107 [2] | JHMDB [18] | YoutubeVIS [52] |
| Metrics | AUC | $\mathcal{J}$-mean | IDF1 MOTA | IDF1 sMOTA | IDF1 MOTA ID-switch (IDs) | PCK | mAP |

A single run of the evaluation on five tasks takes about 2 hours in a Titan Xp GPU.

## B  *Propagation*

### B.1  Box Propagation

In order to propagate bounding boxes, we adopt two methods relying on fully-convolutional Siamese [5, 40, 45, 23] networks. Given a target image patch $I_x$ that contains the object of interest, and a search image patch $I_z$ (typically a larger search area in the next frame), the appearance model $\phi$ processes both patches and outputs their feature maps $x = \phi(I_x)$ and $z = \phi(I_z)$.

**Cross-correlation (XCorr) head.** As in SiamFC [5], we simply cross-correlate the two feature maps, yielding the response map

$$g(x, z) = x \star z \tag{1}$$

Eq. 1 is equivalent to performing an exhaustive search of the pattern $x$ over the search region $z$. The location of the target object can be determined by finding the maximum value of response map.

**Discriminative Correlation Filter (DCF) head.** The DCF head [40, 45] is similar to the XCorr head, with two major differences. The first one is that it involves solving a ridge-regression problem to find the template $w = \omega(x)$ rather than using the original template $x$, so that the response map is given by

$$g(x, z) = \omega(x) \star z \tag{2}$$

More specifically, the DCF template $w = \omega(x)$ is a more discriminative template compared with the original template, and is obtained by solving

$$\arg\min_{w} \|w \star x - y\|^2 + \lambda\|w\|^2, \qquad (3)$$

where $y$ is an ideal response (here represented as a Gaussian function peaked at the center) and $\lambda \geq 0$ is the regularization coefficient typical of ridge regression. The solution to Eq. 3 can be computed efficiently in the Fourier domain [40, 45] as

$$\hat{w} = \frac{\hat{x} \odot \hat{y}^*}{\hat{x} \odot \hat{x}^* + \lambda} \qquad (4)$$

where the hat notation $\hat{x} = \mathcal{F}(x)$ indicates the discrete Fourier Transform of $x$, $y^*$ represents the complex conjugate of $y$ and $\odot$ denotes the Hadamard (element-wise) product. The response map can be computed via inverse Fourier Transform $\mathcal{F}^{-1}$,

$$g(x, z) = \hat{w} \star z = \mathcal{F}^{-1}(\hat{w} \odot z) \qquad (5)$$

Another difference *w.r.t* the XCorr head is that it is effective to update the template online by simple moving average [45], *i.e.* , $\hat{w}_t = \frac{\alpha \hat{x}_t \odot \hat{y}^* + (1-\alpha)\hat{x}_{t-1} \odot \hat{y}^*}{\alpha(\hat{x}_t \odot \hat{x}_t^* + \lambda) + (1-\alpha)(\hat{x}_{t-1} \odot \hat{x}_{t-1}^* + \lambda)}$. In contrast, with the XCorr head every frame is compared against the first one.

As shown in Table 2 and Table 3 from the main paper, for the tested architectures and appearance models we can see a clear advantage of DCF of XCorr (note that the difference was less significant in the original [40] paper, though the experiments were done with a shallower architecture).

**Hyper-parameters.** Following common practice [5, 23], we provide the Correlation Filter with a larger region of context in the template patch. To be specific, the template patch $I_x$ is determined by expanding the height and width of the target bounding box by $k = 4.5$ times. The search patch is also determined by expanding the bounding box by same amount, and its center corresponds the latest estimated location of the target. To handle scale variation of the object, we consider $s = 3$ different search patches at different scales $0.985^{\{1,0,1\}}$. Template and search patches are cropped and resized to $520 \times 520$. This means that with a total stride of $r = 8$, we have feature maps of size $65 \times 65$. In the DCF head, we set the regularization coefficient to $\lambda = 1e^{-4}$, and the moving average momentum to $\alpha = 1e^{-2}$.

| Box prop. hyper-parameters | Values |
|---|---|
| Template patch size | $512 \times 512$ |
| Search patch size | $512 \times 512$ |
| Box expanding coefficient | 4.5 |
| # Scales $s$ | 3 |
| Scale factors | $1.0275^{\{-1,0,1\}}$ |
| Scale penalties | $0.985^{\{1,0,1\}}$ |
| Regularization coefficient $\lambda$ | $1e^{-4}$ |
| Moving average momentum $\alpha$ | $1e^{-2}$ |

### B.2    Mask and Pose Propagation

In Section 2.3 we introduced the recursive mask propagation as $z_t = K_{t-1}^t z_{t-1}$. In practice, to provide more temporal context, we use a memory bank [21, 17] consisting of multiple former label maps as the source label $z_m$ instead of a single label map $z_{t-1}$, *i.e.* $z_t = K_m^t z_m$. More specifically, the resulting source label map is obtained by concatenating all the label maps inside the memory bank, $z_m \in [0, 1]^{Ms}$, where $s$ is the spatial size of a single label map and $M$ is the size of the memory bank. The softmax computed for $K_m^t$ is applied over all $Ms$ points in the memory bank. The memory bank includes the first frame of the video, together with the latest $M - 1$ frames, and we choose $M = 6$. As suggested by MAST [21] and CRW [17], we also introduce the local attention technique, which restricts the source points considered for each target point to a local circle with radius $r = 12$. The hyper-parameter $k$ for the $k$-NN used when computing the transition matrix $K_m^t$ is set to $k = 10$.

Propagating pose key points is cast as propagating the mask of each individual key point, represented with the widely adopted Gaussian belief maps [48]. Each Gaussian has mean equal to the corresponding keypoint's location, and variance proportional to the subject's body size $\sigma = max(\eta s_{body}, 0.5)$. The body size is determined by,

$$s_{body} = \max(\max_p\{x_p\} - \min_p\{x_p\}, \max_p\{y_p\} - \min_p\{y_p\}) \tag{6}$$

where $(x_p, y_p)$ are the coordinates of the $p$-th key point.

| Mask/Pose prop. hyper-parameters | Values |
|---|---|
| Image size | Mask: $480 \times 640$ |
| | Pose: $320 \times 320$ |
| Softmax temperature $\tau$ | 0.05 |
| Memory size $M$ | 6 |
| Local attention radius $r$ | 12 |
| $k$ for $k$-nearest neighbor | 10 |
| Gaussian variance coefficient $\eta$ | 0.01 |

## C  *Association*

### C.1  Association Algorithm

**Motion cues: object states and Kalman Filtering.** We employ a Kalman filter with constant velocity and linear motion model to handle motion cues in algorithms of the *association* type. We assume a generic setting where the camera is not calibrated and the ego-motion is not known. The object *states* are defined in an eight-dimensional space $(u, v, \gamma, h, \dot{u}, \dot{v}, \dot{\gamma}, \dot{h})$, where $(u, v)$ indicate the position bounding box center, $h$ the bounding-box height and $\gamma = \frac{h}{w}$ the aspect ratio. The latter four dimensions represent the respective velocities of the first four terms.

For the sake of simplicity we convert mask representations to bounding boxes. Let the coordinates of "in-mask" pixels form a set $\{(x_j, y_j)|j = 1, ...N\}$, where $N$ is the number of mask pixels. Then, the center of the corresponding bounding box is obtained by averaging these coordinates, as $(u, v) = \frac{1}{N}\sum_{j=1}^{N}(x_j, y_j)$. We estimate the height of the bounding box as $h = \frac{2}{N}\sum_{j=1}^{N}\|y_j - h\|_1$. This estimation is analogous to the one suggested in the continuous case [25]. Consider a rectangle with scale $(2w, 2h)$ whose center locates at the origin of a 2D coordinate plane; by integrating over the points inside of the rectangle, we have $\frac{1}{h}\int_{-h}^{h}\|y\|_1 dy = \frac{2}{h}\int_0^h y dy = h$. For objects represented as a pose, we first convert pose keypoints to masks following Appendix C.2, and then convert masks to boxes.

For each timestep, the Kalman Filter [19] predicts current states of existing tracklets. If a new detection is associated to a tracklet, then the state of the detection is used to update the tracklet state. If a tracklet is not associated with any detection, its state is simply predicted without correction.

We use the (squared) Mahalanobis distance [49] to measure the "motion distance" between a newly arrived detection and an existing tracklet. Let us project the state distribution of the $i$-th tracklet into the measurement space and denote mean and covariance as $\boldsymbol{\mu}_i$ and $\boldsymbol{\Sigma}_i$, respectively. Then, the *motion distance* is given by

$$c_{i,j}^m = (\boldsymbol{o}_j - \boldsymbol{\mu}_i)^\top \boldsymbol{\Sigma}^{-1}(\boldsymbol{o}_j - \boldsymbol{\mu}_i) \tag{7}$$

where $\boldsymbol{o}_j$ indicates the observed (4D) state of the $j$-th detection. We observe that the Mahalanobis distance consistently outperforms Euclidean distance and IOU distance, likely thanks to the consideration of state estimation uncertainty. Using this metric also allows us to filter out unlikely matches by simply thresholding at $95\%$ confidence interval [49]. We denote the filtering with an indicator function

$$b_{i,j} = \mathbb{1}[c_{i,j}^m > \eta]. \tag{8}$$

The threshold $\eta$ can be computed from the inverse $\mathcal{X}^2$ distribution. In our case the degrees of freedom of the $\mathcal{X}^2$ distribution is 4, so the threshold $\eta = 9.4877$.

**Algorithm 1:** Hungarian Association

**Input:** Tracklet indices $\mathcal{T} = \{1, ..., N\}$, detection indices $\mathcal{D} = \{1, ..., M\}$. Hyperparameter $\lambda$.
**Output:** Set of matches $\mathcal{M}$, set of unmatched tracklets $\mathcal{T}_{remain}$, and detections $\mathcal{D}_{remain}$

**1** Initialization: $\mathcal{M} \leftarrow \emptyset, \mathcal{D}_{remain} \leftarrow \mathcal{D}, \mathcal{T}_{remain} \leftarrow \mathcal{T}$ ;
**2 for** $t \in \mathcal{T}$ **do**
**3** $\quad$ Predict the state of the $t$-th tracklet using Kalman Filter
**4 end**
$\quad$ // main matching stage
**5** Compute motion cost matrix $\boldsymbol{C}^m = [c^m_{i,j}]$ using Eq. 7;
**6** Compute appearance cost matrix $\boldsymbol{C}^a = [c^a_{i,j}]$ using Eq. 9;
**7** Compute final cost matrix $\boldsymbol{C} = \lambda\boldsymbol{C}^a + (1 - \lambda)\boldsymbol{C}^m$;
**8** Compute gating matrix $\boldsymbol{B} = [b_{i,j}]$ using Eq. 8;
**9** $[x_{i,j}]$ = Hungarian_assignment $(\boldsymbol{C})$;
**10** $\mathcal{M} \leftarrow \mathcal{M} \cup \{(i,j)|b_{i,j} \cdot x_{i,j} > 0\}$ ;
**11** $\mathcal{T}_{remain} \leftarrow \mathcal{T} \setminus \{i| \sum_j b_{i,j} \cdot x_{i,j} > 0\}$ ;
**12** $\mathcal{D}_{remain} \leftarrow \mathcal{D} \setminus \{j| \sum_i b_{i,j} \cdot x_{i,j} > 0\}$ ;
$\quad$ // second matching stage
**13** Compute IOU cost matrix $\boldsymbol{C}^g$ between $\mathcal{T}_{remain}$ and $\mathcal{D}_{remain}$.;
**14** $[x_{i,j}]$ = Hungarian_assignment $(\boldsymbol{C})$;
**15** $\mathcal{M} \leftarrow \mathcal{M} \cup \{(i,j)|x_{i,j} > 0\}$ ;
**16** $\mathcal{T}_{remain} \leftarrow \mathcal{T}_{remain} \setminus \{i| \sum_j x_{i,j} > 0\}$ ;
**17** $\mathcal{D}_{remain} \leftarrow \mathcal{D}_{remain} \setminus \{j| \sum_i x_{i,j} > 0\}$ ;

**Association algorithm.** Algorithm 1 outlines the association procedure for a *single timestamp*. The algorithm takes as input a set of tracklets $\mathcal{T} = \{1, ..., N\}$ and detections $\mathcal{D} = \{1, ..., M\}$. First, we predict the current states of the all tracklets using the Kalman Filter. Then we perform the main matching stage. In this stage, we compute a motion cost matrix $\boldsymbol{C}^m$ using Eq 7, and compute an appearance cost matrix $\boldsymbol{C}^a$ using the RSM metric described in Section 2.4,

$$c^a_{i,j} = \texttt{RSM}(i, j) \tag{9}$$

The final cost matrix is the linear combination of the two cost matrices $\boldsymbol{C} = \lambda\boldsymbol{C}^a + (1 - \lambda)\boldsymbol{C}^m$. We set $\lambda = 0.99$. A Hungarian solver takes the cost matrix $\boldsymbol{C}$ as input and outputs matches $[x_{i,j}]$. We then filter out unrealistic matches using Eq 8. For the remaining tracklets and detections which failed matching, we perform a second matching stage using IOU distance as the cost matrix. Remaining tracklets and detections are output by the association algorithm, further steps (described below) determine if a remaining tracklet should be terminated or if a new identity should be initialized from a remaining detection.

**Tracklet termination and initialization.** If a tracklet fails to be matched with a newly arrived detection with Algorithm 1, we mark it as inactive. To account for short occlusions, inactive tracklets can still be restored if they are found to be matching with a new detection. We record a "lost age" for each inactive tracklet. If the lost age is greater than a pre-given time, the tracklet would be removed from the current tracklet pool. The lost age is set to 1 second in our experiments.

If a detection fails to match existing tracklets with Algorithm 1, it could correspond to a new tracklet. However, this would result in the creation of frequent brief "spurious" tracklets, containing one detection only. To cope with this issue, similarly to [49] we only initialize a new tracklet if a new detection appears in two consecutive frames (and the IOU between consecutive boxes is at least 0.8).

### C.2 Pose-to-Mask Conversion

Given the key points' location of a target person, we convert the pose into a binary mask in two steps. First, the key points are connected to form a skeleton, where the width of each segment forming this skeleton is proportional to the body size with a linear coefficient $\eta_p = 0.05$, and the

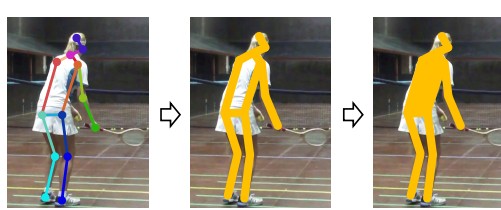

| Detector | DPM | | | FRCNN | | | SDP | | | FairMOT-Det | | | w/ motion |
|---|---|---|---|---|---|---|---|---|---|---|---|---|---|
| Metrics | IDF1 | IDs | MOTA | IDF1 | IDs | MOTA | IDF1 | IDs | MOTA | IDF1 | IDs | MOTA | |
| CF | 36.5 | 748 | 29.8 | 51.3 | 480 | **50.2** | 60.9 | 848 | 64.5 | 75.5 | 550 | **82.9** | ✓ |
| GPF | 34.4 | 1261 | 29.3 | 50.0 | 530 | **50.2** | 60.8 | 985 | **64.8** | 76.4 | 534 | **82.9** | ✓ |
| GF | 36.3 | 674 | 29.8 | 52.2 | 479 | **50.2** | 62.0 | **759** | 64.6 | 75.9 | **499** | **82.9** | ✓ |
| ReID | **40.0** | 619 | 29.8 | 54.9 | 461 | 50.1 | **67.1** | 811 | 64.5 | 78.4 | 545 | 82.8 | ✓ |
| RSM | 39.6 | **513** | **30.0** | **55.6** | **431** | **50.2** | 64.2 | 762 | 64.5 | **78.6** | 543 | 82.7 | ✓ |
| CF | 26.3 | 1381 | 22.8 | 40.4 | 820 | 47.4 | 46.7 | 1525 | 58.2 | 60.4 | 1599 | 76.6 | |
| GPF | 29.5 | 782 | 25.5 | 43.7 | 517 | 48.2 | 48.8 | 1337 | 59.5 | 57.2 | 1414 | 77.4 | |
| GF | 24.9 | 1298 | 22.4 | 41.7 | 526 | 48.1 | 51.0 | **960** | 60.6 | 65.3 | 868 | 78.9 | |
| ReID | **33.1** | **637** | **25.9** | 47.0 | 692 | 47.5 | 53.3 | 1250 | 58.5 | 64.8 | 1448 | 75.9 | |
| RSM | 28.1 | 805 | 25.4 | **51.5** | **414** | 49.8 | **58.6** | 999 | **62.7** | **74.5** | **605** | **82.3** | |

**Table 1:** Comparison between different similarity metrics for association, tested on MOT-16 *train* split. We provide results that (1) use motion cues and (2) discard motion cues. The best results are **bolded** and the second best results are underlined.

| Methods | IDF1 | IDs | MOTA |
|---|---|---|---|
| CF | 38.6 | 6384 | **41.8** |
| GPF | 38.3 | 6245 | **41.8** |
| GF | 39.3 | 5858 | **41.8** |
| ReID | 39.1 | 6442 | 41.7 |
| RSM | **41.3** | **5552** | 41.6 |

**Table 2:** Comparison between different similarity metrics for association, tested on MOT-20 [9] with the provided detector.

| Methods | IDF1 | IDs | sMOTSA |
|---|---|---|---|
| CF | 62.8 | 1529 | 80.7 |
| GPF | 60.7 | 1071 | 82.4 |
| RSM | **66.5** | **808** | **83.4** |

**Table 3:** Comparison between different similarity metrics for *association*, tested on MOTS [43] *train* split based on the segmentation masks provided by the COSTA$_{st}$ [1] tracker.

body size is computed with Eq. 6. Second, we fill closed polygons inside the pose skeleton, since the parts inside the polygon usually belong to the target object.

# D  Ablations for the Reconstruction Similarity Metric (RSM)

In Section 2.4 we claimed that the good tracking performance of UniTrack on association-type tasks is largely attributed to the proposed Reconstruction Similarity Metric (RSM). In this section, we provide results of several baseline methods in order to validate the effectiveness of RSM. These baseline are described below.

**Center feature (CF).** For a given observation feature $d_j \in \mathbb{R}^{s_{d_j} \times C}$ of a bounding box or a mask, we compute the location of its center of mass and extract the corresponding point feature (a single $C$-dim vector) as representation of this observation. Cosine similarity is computed to measure how likely two observations belong to the same identity. Using center feature to represent an object is a straightforward strategy, widely used in tracking tasks [58, 47, 55]. The benefit of CF is that it can handle objects in any observation format, *e.g.* boxes or masks, while the drawback is also obvious: it is a local feature and cannot represent the complete information of the object.

**Global feature (GF).** For a given observation feature $d_j \in \mathbb{R}^{s_{d_j} \times C}$, we concatenate the $s_{d_j}$ point features and obtain a single global feature vector with length $s_{d_j}C$. Cosine similarity is computed to measure how likely two observations belong to the same identity. Note that only representations with fixed $s_{d_j}$ are feasible in this case. For this reason, we only provide results for GF on the MOT task, where observations are bounding boxes that can be resized to a fixed size. The benefit of GF is that it preserve complete information of the observation, while the main drawback is that local features may not align between a pair of samples. Therefore, global feature is only applicable in cases where samples are aligned with pre-processing, *e.g.* in face recognition [13]

**Global-pooled feature (GPF).** Similar to the global feature, but averaging is performed along the $s_{d_j}$ dimension to obtain a single feature vector with length $C$. Cosine similarity then is computed to measure how likely it is that the two observations belong to the same identity. A large body of re-identification (ReID) approaches [38, 59, 37] employ global-pooled feature (on fully supervised learned feature maps). The benefit and drawback are similar to center feature.

| Methods | PCK@0.1 | PCK@0.2 |
|---|---|---|
| TimeCycle [46] | 57.3 | 78.1 |
| UVC [25] | 58.6 | 79.6 |
| CRW [17] | **59.0** | **83.2** |
| I18 (reported in [17]) | 53.8 | 74.6 |
| I18 (UniTrack) | 58.3 | 80.5 |
| Yang et al. [53] | **68.7** | **92.1** |

**Table 4:** Results of pose propagation on JH-MDB [18] dataset. I18 refers to using ImageNet pre-trained ResNet-18 as the appearance model.

| Methods | mAP↑ |
|---|---|
| FEELVOS [42] | 26.9 |
| SipMask [6] | **32.5** |
| OSMN† [54] | 27.5 |
| DeepSORT† [49] | 26.1 |
| MTRCNN† [52] | 30.3 |
| UniTrack† | 30.1 |

**Table 5:** VIS results@YoutubeVIS [52] *val* split. † indicates methods using the same observations (segmentation masks in every single frames).

**Supervised ReID feature (ReID).** For a given image cropped from a bounding box, we employ an strong, off-the-shelf person ReID model to extract a single feature vector with length $C$, and compute cosine similarity between observations. The model uses a ResNet-50 [15] architecture and is trained with the joint set of three widely-used datasets: Market-1501 [56], CUHK-03 [24], and DukeMTMC-ReID [34]. Using supervised ReID models to extract appearance features is widely used in existing multi-object tracking approaches [39, 27, 35]. Considering large amount of identity labels are leveraged in training, supervised ReID models usually show good association accuracy.

Note that for CF, GF, GPF, and the proposed RSM, we employ the same appearance model (ImageNet pre-trained ResNet-18) for fair comparison. For a broad comparison, we provide results obtained with different detectors and on different datasets. We adopt the following detectors and test on MOT-16 [29] *train* split (listed with detection accuracy from low to high): DPM [14], Faster R-CNN [33] (FRCNN), SDP [51], and FairMOT [55].

Results are shown in Table 1. We first apply the full association algorithm, *i.e.* using both appearance and motion cues. In this case (first half of the table), RSM consistently outperforms CF, GF, GPF baselines, and even surpasses the supervised ReID features in several cases, *e.g.* with FRCNN and FairMOT detectors. In the second half of the table, we show results in which only appearance cues are used, so that the difference between metrics (which are based on appearance) can be better emphasized. In this case, the gaps between different methods are more significant than in the previous case, and RSM still consistently outperforms CF, GF, and GPF. Furthermore, RSM also surpasses the strong supervised ReID feature with all detectors, except for DPM. This suggests that RSM can be an effective similarity metric for tasks that have *association* at their core.

To show the generality of the results, we also experiment on different datasets and different tasks. Table 2 shows comparisons on the MOT-20 [9] *train* split for the MOT task (box observations). The MOT-20 dataset is specialized for the extreme crowded person tracking scenario. Table 3 presents results on MOTS [43] *train* split for the MOTS task (mask observations). Note for the MOTS task, since the observations (masks) vary in size, it is not feasible to apply the GF strategy. Results show that the proposed RSM yields significantly higher IDF1 scores on both datasets.

## E   More Tracking Tasks

In this section we present two more tasks that UniTrack can address.

The first task is human **Pose Propagation** on the JHMDB [18] dataset: each video contains a *single* person of interest, and the pose keypoints are provided in the first frame of the video only. The goal here is to predict the pose of the person throughout the video. Note that this is different from the previously mentioned PoseTrack task: PoseTrack mainly focuses on association between different identities, while in Pose Propagation we aim at propagating the pose of a single identity.

Results are shown in Table 4. We report a higher result with ImageNet pre-trained ResNet-18 compared with in previous work [17, 25] (58.3 *v.s.* 53.8 PCK@0.1). With this result, we observe the best self-supervised method CRW [17] does not beat the ImageNet pre-trained representation by a significant margin (only +0.7 PCK@1). This again validates our second finding in Section 3.2: a vanilla ImageNet-trained representation is surprisingly effective.

| Method | TS sup. | TrackingNet [31] | | TC-128 [26] | | TLP [30] | | LaSOT [12] | | OxUvA [41] |
|---|---|---|---|---|---|---|---|---|---|---|
| | | Succ. | Prec. | Succ. | Prec. | Succ. | Prec. | Succ. | Prec. | MaxGM |
| KCF [16] | N | 41.9 | 44.7 | 38.7 | 54.9 | 8.4 | 6.3 | 17.8 | - | - |
| ECO [7] | N | 56.1 | 48.9 | - | - | 20.2 | 21.2 | 32.4 | 30.1 | 0.314 |
| Staple [4] | N | - | - | - | - | - | - | - | - | 0.261 |
| BACF [20] | N | - | - | - | - | - | - | - | - | 0.281 |
| SiamFC [5] | Y | 57.1 | 66.3 | 50.3 | 68.8 | 23.5 | 28.4 | 33.6 | 33.9 | 0.313 |
| CFNet [40] | Y | 53.3 | 57.8 | - | - | - | - | 27.5 | - | - |
| SiamRPN [23] | Y | - | - | - | - | - | - | - | - | - |
| SiamRPN++ [22] | Y | 73.3 | 69.4 | - | - | - | - | 49.6 | 49.1 | - |
| LUDT [44] | N | 46.9 | 54.3 | 51.5 | 67.1 | - | - | 26.2 | - | - |
| LUDT+ [44] | N | 49.5 | 56.3 | 55.2 | 72.5 | - | - | 30.5 | - | - |
| UnTrack | N | 59.1 | 51.2 | 54.5 | 73.1 | 25.4 | 23.2 | 35.1 | 32.6 | 0.334 |

**Table 6:** Results on more SOT datasets. An ImageNet pre-trained representation with a ResNet-50 architecture is employed as the appearance model within UniTrack. "TS sup." indicates whether the method requires task-specific supervision.

The second task is **Video Instance Segmentation (VIS)**. The problem of VIS is similar to Multiple Object Tracking and Segmentation (MOTS), but its setup differs in the following aspects: first, the object categories are fairly diverse (40 different categories), while in MOTS objects are mostly persons and vehicles. This also requires the trackers tackling the VIS task to handle objects from different classes within the same scene. Second, the evaluation metrics are different. In MOTS, the MOT-like metrics (CLEAR [3], IDF-1/IDs, and HOTA [28]) are used, which implicitly encourages methods to focus on outputting temporally consistent trajectories. Instead, for VIS the evaluation metric is spatial-temporal mAP, a temporal extension of the vanilla mAP which is usually used in detection and segmentation tasks. The mAP metric significantly biases towards segmentation and classification accuracy in single frames, thus being less informative for evaluating "tracking" accuracy.

Results on VIS task are shown in Table 5. We adopt an identical segmentation model to the one of MaskTrackRCNN [52], and observe only a 0.2 difference in mAP. For further comparison, we also provide results of two other association methods, OSMN [54] and DeepSORT [49], providing them with the same observations as used by UniTrack. Note how UniTrack boasts better accuracy than both methods (30.0 *v.s.* 27.5 and 26.1 mAP). Comparing with a state-of-the-art model, SipMask [6], our result is also comparable with $-2.4$ point mAP. We believe if equipped with more advanced single frame segmentation model, the mAP would be further improved.

## F   SOT results on more datasets

To further validate the general validity of our experiments, we provide more results for the SOT task by testing on more recent datasets that contain large-scale and long-term videos.

The results in Table 6 show a very similar trend to the one already observed for OTB (Table 3e in the main text): For the SOT task, UniTrack with ImageNet features has comparable performance to the one of the recent LUDT+, which like UniTrack does not require task-specific supervision, but can only be used for SOT. Again, similarly to what was reported for OTB, UniTrack is outperformed by recent methods such as SiamRPN++. This is to be expected, as SiamRPN++ is specifically designed for SOT and trained in a supervised fashion on several large-scale video datasets.

## G   Additional Correlation Studies

In Section 3.3 (main paper) we investigated the correlation between tracking performance and ImageNet "linear probe" accuracy for different SSL models. In this section, we provide more results and discussions by studying the correlation between tracking performance and several other downstream tasks when using the appearance model from the many SSL methods under consideration. For non-tracking tasks, we report numbers from [10] and plot them against tracking performance in Figure 1.

We report three tasks: surface normal estimation on the NYUv2 [36] dataset, where the mean angular error is used as the evaluation metric (the lower the better); Object detection on Pascal VOC [11],

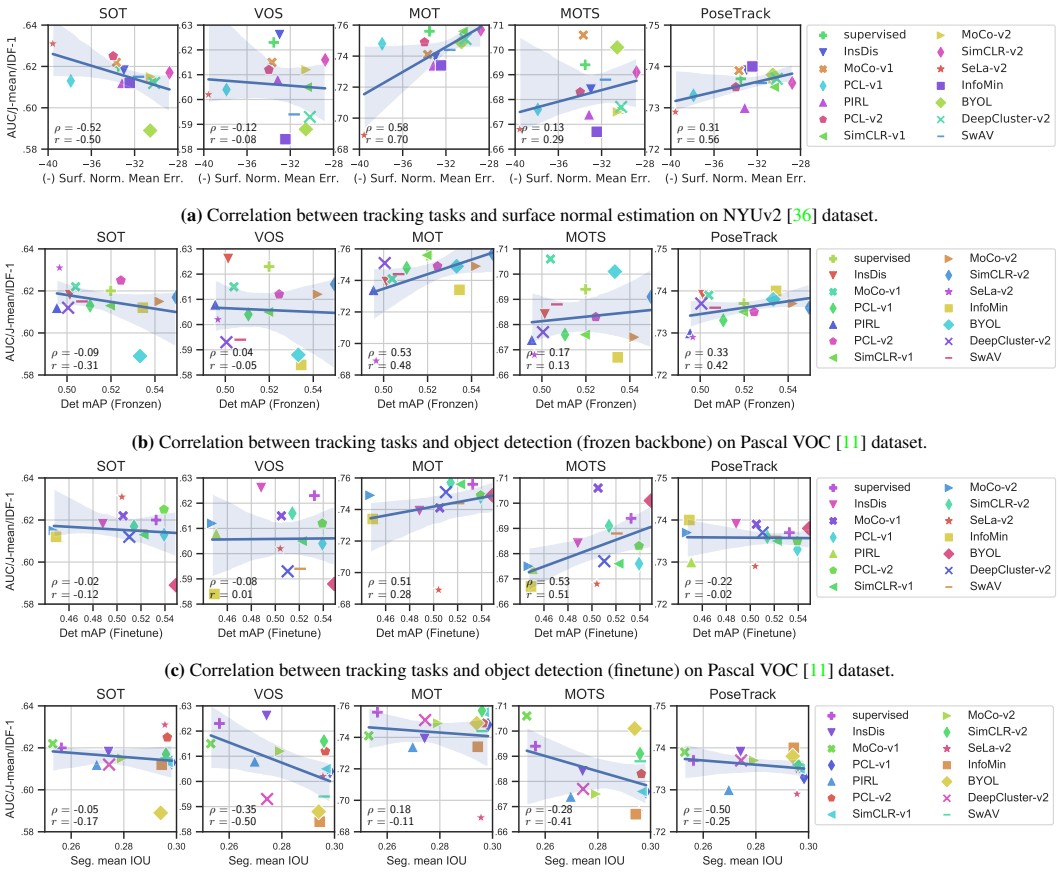

**(a)** Correlation between tracking tasks and surface normal estimation on NYUv2 [36] dataset.

**(b)** Correlation between tracking tasks and object detection (frozen backbone) on Pascal VOC [11] dataset.

**(c)** Correlation between tracking tasks and object detection (finetune) on Pascal VOC [11] dataset.

**(d)** Correlation between tracking tasks and semantic segmentation on ADE20k [57] dataset.

**Figure 1:** Correlation study between tracking tasks and other tasks for SSL models. On the y-axes we plot tracking performance, and on x-axes performance of the other tasks. Spearman's $r$ and Pearson's $\rho$ are shown in the left bottom corner of each plot, indicating how the two axes are correlated.

with performance measured in mAP (the higher the better); Semantic segmentation on ADE20k [57] dataset, with performance measured in mean IOU (the higher the better). In each subfigure, we plot the performance of five tracking tasks along the y-axes, and performance of the other task along the x-axes. Note that we actually use *negative* mean error for surface normal estimation, to represent *accuracy*. As in the main paper, we compute two types of correlation coefficient: Spearman' $r$ and Pearson's $\rho$, and report them in the left bottom corner of each plot. Several interesting findings can be observed:

*(a) Correlation between tracking and surface normal prediction performance is fairly strong.* Results are shown in Figure 1a. For instance, $r = 0.70$ for surface normal error *v.s.* MOT accuracy, and $0.56$ for surface normal error *v.s.* PoseTrack accuracy. Interestingly, the behavior of SOT is in contrast with MOT and PoseTrack: SOT accuracy is moderately negative correlated ($r = -0.50$) with surface normal estimation accuracy. VOS presents a similar trend to the one of SOT, but with a lower correlation coefficient.

*(b) Object detection is moderately correlated with association-type tracking tasks.* For object detection, we consider two setups: one is to freeze the representation and only train the additional classification/regression head; the other is to finetune the whole network in an end-to-end manner. Results are shown in Figure 1b and 1c respectively. In general, MOT and PoseTrack are moderately correlated with object detection under the frozen setting ($r = 0.48$ for MOT and and $r = 0.42$ for PoseTrack), and MOTS is moderately correlated with object detection under the finetune setting ($r = 0.51$). Propagation-type tasks are poorly correlated with object detection results under both settings ($|\rho| < 0.10$). We speculate that, in this case, positive correlation might be due to the fact that

both object detection and association-type tracking require discriminative features at the level of the object.

*(c) Semantic segmentation is slightly negative correlated with tracking tasks.* As can be observed in Figure 1d, correlation coefficients between segmentation accuracy and tracking performance are mildly negative. Among these results, VOS is the task that is most (negatively) correlated with segmentation, with $r = -0.50$. MOTS and PoseTrack are also mildly correlated, with $r = -0.41$ and $r = -0.25$ respectively. We speculate that negative correlation might be cause to the fact that tracking and segmentation require features with contradictory properties. Consider two different instances that belongs to the same category, *i.e.* two different pedestrian. For segmentation, the task requires pixel-wise classification, meaning that pixels inside the two instances should be equally classified into the same "pedestrian" class, thus their features should be *similar* (close to the class center). In contrast, for tracking tasks, it is required to distinguish different instances from the same class, otherwise a tracker would easily fail when objects overlap with each other. Therefore, point features inside the two different pedestrian are expected to be *dissimilar*.