# OpenReview forum: "Do Different Tracking Tasks Require Different Appearance Models?"
_NeurIPS.cc/2021/Conference — NeurIPS 2021 Poster_

### Official Review · Reviewer_KcZZ · 2021-07-14

**Rating:** 7
**Confidence:** 3

**Summary:**

This paper categorizes tracking object tasks into five lines and divides tracking algorithms into two classes: propagation and association. It indicates no necessity to fragment the problem of tracking in so many different specifications and separate trained models for each one. A unified tracking solution is proposed to address five different tasks using the same appearance model, which is task-agnostic and can be learned in a supervised or self-supervised way. The proposed model contains multiple heads to address individual tasks without training, and a reconstruction-based similarity metric is designed for association that preserves fine-grained visual features and supports multiple observation formats (box, mask and pose). I feel this work is interesting and has strong contributions to the tracking community, but experimental results are somewhat weak.

**Limitations And Societal Impact:**

Yes.

**Main Review:**

Strengths

-- The method is novel. It should be the first simple and universal framework that simultaneously supports five tracking tasks (SOT, VOS, MOT, MOTS, PoseTrack), and can also be easily extended to new ones.

-- Compared with traditional cosine similarity methods, which do not support varying spatial sizes, this work proposes a reconstruction-based similarity metric that is able to effectively deal with different observation formats while preserving fine-grained information.

-- The framework has achieved competitive performance against dedicated methods on the five tasks of SOT, VOS, MOT, MOTS and PoseTrack. Moreover, unlike existing self-supervised learning methods that require additional training, the method proposed in this paper does not require additional training for different tasks.

Weaknesses

-- This framework does not train an appearance model on a single target task, nor does it apply multi-task learning on multiple tasks. The complementary benefits of different tracking tasks are not explored and exploited, since these tracking tasks have some common properties that can be utilized for the enhancement to each other. It would be interesting if authors perform shared representation learning across different tasks and fine-tune task heads to consider their specialty, although domain difference and optimization object conflict might exist.

--UniTrack performs comparable with SOTA methods on MOT, MOTS and poseTrack, but greatly worse on SOT and VOS. Moreover, some settings make comparison unconvincing. For example, in the task of SOT, comparison is only made on a small-scale and somewhat out-of-date dataset and recent SOTA methods are missing.


**Time Spent Reviewing:**

Five hours

---

> ### Author Response · Authors · 2021-08-10
> **Response to reviewer KcZZ**
>
> We thank reviewer KcZZ for their time and constructive feedback. We are glad to hear that they found this work _“interesting and a strong contribution to the tracking community”_.
>
> ----
> **1.**
> > _“It would be interesting if authors perform shared representation learning across different tasks and fine-tune task heads to consider their specialty, although domain difference and optimization object conflict might exist.”_
>
> We agree this is an interesting direction - extending UniTrack to work in a multi-task framework could lead to an improvement across the board, as the appearance model could potentially more expressively encode a good shared representation among the _specific_ set of tasks of interest.
> As a first iteration, one could use an imagenet-trained or the latest self-supervised model as a strong initialisation of the appearance model (since we proved them being effective) and then experiment with training on a subset of the several tasks considered in a multi-task fashion.
> Despite the appeal, this requires a notable amount of extra work and experiments. For instance, all the task-specific heads need to be made differentiable, per-head losses need to be designed and the project code significantly adapted (starting from the data loader).
> Hence, we consider experiments on a multi-task configuration a promising direction for future work that can be informed by the insights presented in our paper.
> We will extend the paper to explicitly suggest multi-task learning as a natural future development.
>
> ----
> **2.**
> > [Comparable with state of the art for MOT, MOTS and PoseTrack, worse for SOT and VOS]
>
> We agree that, from a performance perspective, the use of UniTrack is mostly convenient for MOT, MOTS and PoseTrack and less so for SOT and VOS, which presents a gap wrt the most recent literature.
> However, please note that the purpose of our work is not to be strongly competitive across all the tasks considered. In fact, we were surprised to see that many pretrained features used in UniTrack could reach such a high performance in MOT, MOTS and Posetrack. Instead, our two primary goals are a) understanding whether similar tracking tasks can be addressed with the same appearance model and b) evaluate state-of-the-art self-supervised features across different tracking tasks. Nonetheless, this is a good point to make - we will extend our introduction and discussion to better distinguish results on SOT and VOS from those on MOT, MOTS and PoseTrack.
>
> Note also that the results reported in Table 1 mostly serve to contextualize the performance of the vanilla imagenet-trained appearance model, which is the best if we average its performance across tasks, but not always the best choice if we allow ourselves to choose a difference appearance model for every task. For instance, for SOT (DCF head) features from SeLaV2 are +1.3% more effective, and for VOS task CRW features would even grant an extea +4.8% with respect to what reported in Table1 (see Table 2 and 3).
> Moreover, for SOT it is possible to significantly increase the performance with a few modifications described in the paper "SiamFC++: Towards Robust and Accurate Visual Tracking with Target Estimation Guidelines", such as adding a regressor to refine the bounding box prediction at every frame. The set of guidelines introduced in this paper manage to improve over SiamFC of +7.4% absolute points on OTB and 16.5% on LaSOT. Instead,  we opted to keep the framework as simple as possible.
>
> Finally, note how all these methods are dedicated (both in terms of design and training) to a specific task, and often even trained on the same distribution of the dataset on which they are tested. In the light of this, we believe that direct comparisons between UniTrack and the state of the art  should be made with the appropriate caveats.
>
> ----
> **3.**
> > _” in the task of SOT, comparison is only made on a small-scale and somewhat out-of-date dataset and recent SOTA methods are missing.”_
>
> We reported SiamRPN++ as representative of the state of the art for supervised methods, together with its earlier iteration SiamFC and SiamRPN; and LUDT+ as representative for unsupervised methods. We opted for not reporting the very latest single-object tracking methods, as they are significantly more complex, but we are happy to include them in the next update of the manuscript for completeness. Please let us know if you have any suggestion on which ones would be the most appropriate for comparison.
>
> We have also run UniTrack with a vanilla imagenet-trained appearance model on a more up-to-date (and also larger-scale) dataset, LaSOT. Results are reported below and show a similar trend to the one of Table 1a from our paper.
>
> | Method        | Task-specific supervision      | success    | precision     |
> | ---             | ---                 | ---         | ---         |
> |KCF            |n                |17.8        |    -    |
> |CFNet            |y                |27.5        |    -    |
> |ECO            |n                |32.4        |30.1        |
> |SiamFC        |y                | 33.6        | 33.9        |
> |SiamRPN++        |y                |49.6        | 49.1        |
> |LUDT            |n                |26.2        | -        |
> |LUDT+        |n                |30.5        | -        |
> |UniTrack |n                |35.1        | 32.6        |
>
> ----
>
> **4.**
> As a final note: we have tested our framework with the appearance model of the very recent self-supervised method VFS, from the paper “Rethinking Self-Supervised Correspondence Learning: A Video Frame-level Similarity Perspective" [Xu&Wang; ICCV 2021]). We found that it outperforms SimCLRv2 (the best self-supervised model from our submission) on every metric and task except one. For the results, please see the general comment on the top of the thread titled “UniTrack results with VFS features”.

---

### Official Review · Reviewer_7ktv · 2021-07-15

**Rating:** 7
**Confidence:** 4

**Summary:**

The paper investigates the possibility of using a single appearance model for all tracking-related tasks/subtasks (the paper proposes a precise classification of these tasks). The authors propose an elegant unified framework with 3 levels: basic appearance model (level 1), propagation and association blocks (level 2), and task-specific heads (level 3). The only level that needs a training phase is level 1, while the other levels are training-free. Another contribution is the introduction of the RSM metric, which is used to measure the similarity between features within the proposed association algorithm. Following a rich experimental section, the authors conclude by giving a negative answer to the question contained in the title of the paper itself.

**Limitations And Societal Impact:**

Not applicable, as the paper does not have a significant social impact.


**Main Review:**

-**Originality**: The authors propose a method with some innovative elements and an interesting combination of well-known techniques to tackle the problem of object tracking in videos. The main innovative contribution, in my opinion, is the RSM metric, which is used to measure the similarity between features within the proposed association algorithm. The presented framework, as a whole, is interesting, although it takes many techniques and ideas from pre-existing works (which are correctly cited by the authors). Overall, I believe that the originality level of the paper is not high, but still acceptable.
-**Quality**: Technically, the paper is of good quality. The claims of the paper are well supported by the theoretical treatment, but they lose some points due to the experimental results that, in my opinion, are not always amazing. However, the authors highlight the weaknesses of their approach and explain the reasons behind the most underwhelming results. The paper therefore shows how the use of a different appearance model for each tracking subtask is not strictly necessary to have competitive results; but in my opinion it also highlights how a specific approach may in some cases be necessary to obtain SoTA results.
-**Clarity**: The paper is written in a clear and well-organized way. From reading the paper, you have all the information you need to re-implement the method. Furthermore, the supplementary material further enriches the discussion by making the paper even clearer. My only note in this sense is related to the figures: some of them, in particular Figure 3, would in my opinion have benefited from more space and a greater amount of detail; relegating the overall scheme of the framework to a small figure with few details can be a bit limiting (the details are however contained in the paper body).
-**Significance**: Some elements that the paper proposes could be reused by the community, such as the RSM metric and the categorization of the various tracking-related subtasks. The framework as a whole can also have some relevance when used as a baseline for comparison with other methods, particularly for methods that use a specific approach to a given subtask. However, it does not seem to me that the conclusions of the paper are particularly significant, because the experimental results (that are not always excellent) make the authors' main claim a little bit weak.
-**Overall Opinion**: The paper is well done and contains some interesting elements. Not particularly groundbreaking, but acceptable. Given the experimental results, I would propose the framework just as a baseline for task-specific methods, because I don't think the answer to the question that gives the paper its title is a definitive “no”.

**Time Spent Reviewing:**

2.5

---

> ### Author Response · Authors · 2021-08-10
> **Response to reviewer 7ktv**
>
> We would like to thank reviewer 7ktv for their time and constructive comments.
>
> ----
>
> >  _“I don't think the answer to the question that gives the paper its title is a definitive ‘no’.”_
>
> We agree that this should have been discussed better. Despite the competitive performance across the board (especially for the association tasks MOT, MOTS and PoseTrack), a single general-purpose appearance model can not yet beat all the best existing methods tailored on individual specific tasks. The spirit of the rhetorical question that titles the paper was rather to question whether the sharing of features would work reasonably well or not. However, the tone got a bit lost throughout the paper.
> For this reason, we will revise the conclusion of the paper to give it a more conservative tone and discuss the experiments in section 3.1 more extensively to better distinguish results on propagation and association tasks.
>
> What we wanted to convey in our paper is that a unified framework like UniTrack is an appealing alternative to task-specific methods because it allows us to leverage progress in representation learning with no extra cost. For instance, we were able to easily use the features obtained from a very recent self-supervised method (VFS, from _"Rethinking Self-Supervised Correspondence Learning: A Video Frame-level Similarity Perspective"_ [Xu&Wang; ICCV 2021]).
> We found that it outperforms SimCLRv2 (the best self-supervised model from the submission) on almost every metric and task. For the results, please see the general comment on the top of the page titled “UniTrack results with VFS features”.
>
> ---
> > _”Figure 3 would in my opinion have benefited from more space and a greater amount of detail”_
>
> Thank you for pointing this out, we will make it clearer and place it at the top of the page.

---

### Official Review · Reviewer_igQk · 2021-07-16

**Rating:** 4
**Confidence:** 5

**Summary:**

This paper proposes a unified visual tracking method using a single and task-agnostic appearance model, in which this model can be learned via n a supervised or self-supervised strategy. For this purpose, they examine appearance model performance on five tracking tasks and describe it using radar charts in fig1. Experimental results show that  most tracking tasks can be solved within the proposed framework.


**Limitations And Societal Impact:**

All compared methods were not pre-traineded using ImageNet, thus comparisons can be unfair.
The experimental evaluations are insufficient to demonstrate the effectiveness of the proposed unified appearance model under various visual tracking environments. The method needs to be evaluated using single target tracking datasets including OTB, VOT, LaSOT, OxUvA, TLP, TrackingNet, UAV123, TC, and GOT-10k. The compared methods are not recent state-of-the-art methods. More recent methods need be compared. It needs to be proven theoretically that the proposed appearance modeling strategies can cover most changes in target appearance and adapt to change the appearance under a current visual tracking environment.  It would be better to discuss and compare the proposed method against the fusion method of multiple appearance models. The paper also needs to explain why box, mask, and pose representations are considered.

**Main Review:**

The paper considered a unified framework for the appearance model. However it is unclear why the method can contribute improve the visual tracking performance. While visual tracking environments considerably vary over time, visual tracking method needs to be specialized to a current visual tracking environment, rather than using a unified appearance model.

**Time Spent Reviewing:**

7

---

> ### Author Response · Authors · 2021-08-10
> **Response to reviewer igQk**
>
> We thank reviewer igQk for their time and feedback.
> Please note that in this comment we refer to papers with letters. The list of references is given at the end of this response.
>
> ----
>
> **1**.
> First, we would like to address what seems to be the main concern:
> > _"it is unclear why the method can contribute improve the visual tracking performance [... ] visual tracking method needs to be specialized to a current visual tracking environment, rather than using a unified appearance model”_
>
> Thank you for pointing this out - we will revise the text to make this clearer. We agree that designing dedicated algorithms specialised to specific tracking tasks is important, if performance is paramount. However, in this paper we offer a different angle by asking ourselves the question: _can several tracking tasks be addressed within the same framework?_ The aim is not to achieve state-of-the-art results across the board, but rather to uncover what it is that these different tasks have in common. We believe both angles (i.e. both the task-specific approach and our unified framework) are valuable and their usefulness depends on the context.
>
> Nonetheless, the framework we presented also allows us to directly exploit new findings in representation learning and exploit the most recent state of the art without any adaptation. For instance, we were able to immediately re-use the features obtained from a very recent self-supervised method (VFS, from "Rethinking Self-Supervised Correspondence Learning: A Video Frame-level Similarity Perspective" [Xu&Wang; ICCV 2021]). We found that it outperforms (sometimes significantly) the best self-supervised model from the submission SimCLRv2 on every metric and task except one. For the results, please see the general comment on the top of the page titled “UniTrack results with VFS features”. Hopefully this should further demonstrate that our contribution can facilitate the improvement of tracking performance of the field (which is the main concern expressed by this reviewer) by allowing the direct exploitation of novel developments in representation learning.
>
> ----
> **2.**
>
> > _“All compared methods were not pre-trained using ImageNet, thus comparisons can be unfair.”_
>
> Given the large-scale evaluation we conducted, it would be impossible for us to retrain all the methods on exactly the same training data. More importantly, by using a non-video and non-task-specific dataset such as imagenet _the disadvantage is on us_, not on the methods we compare against. Finally, please note that most of the supervised methods we compare against actually initialise their features on an imagenet-trained embedding before training on task-specific datasets.
>
> ----
> **3.**
>
> > _“The method needs to be evaluated using single target tracking datasets including VOT, LaSOT, OxUvA, [...]”_
>
> We extended our results by computing UniTrack’s SOT performance also on LaSOT, which can be found below. With UniTrack we refer to vanilla imagenet features, a DCF head and features from ResNet block3. Note how the trend of Table 1a) is preserved.
>
> | Method        | Task-specific supervision      | success    | precision     |
> | ---             | ---                 | ---         | ---         |
> |KCF            |n                |17.8        |    -    |
> |CFNet            |y                |27.5        |    -    |
> |ECO            |n                |32.4        |30.1        |
> |SiamFC        |y                | 33.6        | 33.9        |
> |SiamRPN++        |y                |49.6        | 49.1        |
> |LUDT            |n                |26.2        | -        |
> |LUDT+        |n                |30.5        | -        |
> |UniTrack  |n                |35.1        | 32.6        |
>
> To further clarify our choice of benchmarks in our submission: Given the large number of experiments already conducted across five tasks (plus two in the Supplementary). We opted for using, for every task, what we found being the most popular benchmark.
>
> ----
> **4.**
> > _“It needs to be proven theoretically that the proposed appearance modeling strategies can cover most changes in target appearance”_
>
> We respectfully disagree with this point. Proving the expressiveness of learned representations is surely interesting (see e.g. [A]), but it’s beyond the scope of this paper, which is a large-scale empirical study that is inductive by design.
>
> Regarding this point, to design our method and our experiments we relied on intuitions built from prior work which have showed that:
> * Mid-level features of imagenet-trained convnet classifiers are actually able to address fine-grained/local correspondences [B]. This is important especially for the _association_ tasks, which require discriminating between similar objects (where local differences matter a lot).
> * Appearance models from state-of-the-art self-supervised approaches “transfer” well to a variety of (image-based) tasks such as few-shot classification, object detection and surface normal estimation [C].
>
> ----
> **5.**
>
> > _“It would be better to discuss and compare the proposed method against the fusion method of multiple appearance models”_
>
>
> We are not sure what the reviewer means with _“the fusion method of multiple appearance models”_. Would it be possible to clarify?
>
>
> ----
> **6.**
> > _“The paper also needs to explain why box, mask, and pose representations are considered.”_
>
>
> We chose these representation formats simply because they are the default choices for the tasks and benchmarks taken into account in our paper: boxes for SOT and MOT, masks for VOS, MOTS (and VIS, in the Supplementary), and pose for PoseTrack. We will add a note in the text to better clarify this.
>
> ----
>
> We hope that our answers satisfy this reviewer concerns. If not, we are happy to engage in further discussion.
>
>
> **References**
> * [A] What Makes for Good Views for ContrastiveLearning?; Tian et al.; NeurIPS 2020
> * [B] Do Convnets Learn Correspondence?; Long et al.; NeuIPS 2014
> * [C] How Well Do Self-Supervised Models Transfer?; Ericsson et al.; CVPR 2021

---

> > ### Comment · Reviewer_igQk · 2021-08-19
> > **The basic idea of the proposed method is interesting, but several concerns raised by us were not carefully  addressed in the responses from authors.**
> >
> > The basic idea of the proposed method is interesting and the method can contribute to improve the visual tracking accuracy.
> > Thus, we raise the score to "Ok but not good enough - rejection".
> > Nevertheless, several concerns raised by us were not carefully  addressed in the responses from authors.
> >
> > 1. The method was evaluated by few datasets for "single" target tracking problems.
> > It is highly recommended that the method needs to be evaluated using OxUvA, TLP, TrackingNet, TC, and GOT-10k.
> >
> > 2. We believe that  many conventional methods can considerably improve their performance, if they can be pre-trained using ImageNet. We agree that it is difficult to retrain all the methods on exactly the same training data. Nevertheless, there should be attempts to  evaluate compared methods under at least similar environments. In addition, the paper need to contain more explanation and analysis on that ImageNet features are not one of  key factors for improving the accuracy.
> >
> > 3. To make the proposed method not to be ad-hoc,  the paper needs to contain theoretical analysis as well as empirical observations. How to theoretically  explain that the proposed appearance modeling strategies can cover most changes in target appearance?

---

> > > ### Author Response · Authors · 2021-08-29
> > > **Response to additional comments**
> > >
> > > We thank reviewer igQk for the additional comments.
> > >
> > > > _“many conventional methods can considerably improve their performance, if they can be pre-trained using ImageNet.”_
> > >
> > > We believe there is a misunderstanding around this point.
> > > Most of the methods we compare against are _already_ pre-trained on ImageNet. For these methods, this is then typically followed by task-specific training on video datasets. These methods include SiamRPN, SiamRPN++, SiamMask, FEELVOS, STM, and _all_ methods in Table 1 (c) (d) (e).
> > > This is what we meant in our previous answer when we said that the “disadvantage is on us”: unlike these methods, we did not rely on task-specific training on dedicated datasets.
> > > Few methods (Colorization, TimeCycle, UVC, CRW, SiamFC) are not pre-trained on ImageNet, but they are still trained on video datasets in a task-specific manner.
> > >
> > > Importantly, please also note that in Table 2 in each column the best self-supervised appearance model always outperforms ImageNet features (reported in row 2) except in one case (column 1).  So if we were to pick the best features for each task for reporting our performance in Table 1, we would overall achieve a more favourable case for UniTrack, across the board. We picked ImageNet-pretrained features for Table 1 just because we wanted to keep the comparison simple.
> > >
> > > > _“How to theoretically explain that the proposed appearance modeling strategies can cover most changes in target appearance?.”_
> > >
> > > We already answered this question in point 4 of our previous answer. Could you kindly let us know with what part of our argument you disagree?
> > >
> > > > Evaluation on more datasets for the single-object tracking task (SOT)
> > >
> > > We extended the evaluation to four further SOT datasets: TrackingNet, OxUvA, TC128, and TLP. These results are in addition to the ones on OTB-15 and LaSOT already reported.
> > > Note that we did not perform such a large scale analysis for SOT on many datasets in our submission only because SOT is just _one_ out of the five tasks considered for UniTrack: reporting results on a total of twenty-five to thirty datasets would have been prohibitive given that we are considering many different features and configurations (see Table 2 and 3, this would have required thousands of runs).
> > >
> > >
> > > The results below show a very similar trend to the one already observed for OTB (Table 1a) and LaSOT (previous answer): For the SOT task, UniTrack with ImageNet features has comparable performance to the one of the recent LUDT+, which like UniTrack does not require task-specific supervision, but can only be used for SOT.
> > > Again, similar to what was reported for OTB and LaSOT, UniTrack is outperformed by recent methods such as SiamRPN++.This is to be expected, as SiamRPN++ is specifically designed for SOT and trained in a supervised fashion on several large-scale video datasets.
> > > (Note that SiamRPN++ and LUDT+ results are not available for some of the datasets requested).
> > >
> > > Nonetheless, for MOT, MOTS, and PoseTrack, UniTrack is actually competitive with state-of-the-art supervised methods that have been trained specifically for the task.
> > > We will improve the discussion to better highlight the difference in performance between MOT/MOTS/PoseTrack and SOT/VOS.
> > >
> > > **TrackingNet test set**
> > >
> > > | Method		| Task-specific supervision 	 | success	| precision 	|
> > > | --- 			| ---		 		| ---	 	| --- 		|
> > > |KCF			|n				|41.9		|44.7		|
> > > |CFNet			|y				|53.3		|57.8		|
> > > |ECO			|n				|56.1		|48.9		|
> > > |SiamFC		|y				|57.1		|66.3		|
> > > |SiamRPN++		|y				|73.3		|69.4		|
> > > |LUDT			|n				|46.9		|54.3		|
> > > |LUDT+		|n				|49.5		|56.3		|
> > > |UniTrack [imagenet features] |n				|59.1		|51.2		|
> > >
> > > **OxUvA**
> > >
> > > | Method		| Task-specific supervision 	 | MaxGM	|
> > > | --- 			| ---		 		| ---	 	|
> > > |Staple			|n				|0.261		|
> > > |BACF			|n				|0.281		|
> > > |ECO			|n				|0.314		|
> > > |SiamFC		|y				|0.313		|
> > > |UniTrack [imagenet features] |n			|0.334		|
> > >
> > > **Temple Color 128 (TC128)**
> > >
> > > | Method		| Task-specific supervision 	 | success	| precision 	|
> > > | --- 			| ---		 		| ---	 	| --- 		|
> > > |KCF			|n				|38.7		|54.9		|
> > > |CFNet			|y				|45.6		|60.7		|
> > > |SiamFC		|y				|50.3		|68.8		|
> > > |LUDT			|n				|51.5		| 67.1		|
> > > |LUDT+		|n				|55.2		| 72.5		|
> > > |UniTrack [imagenet features] |n				|54.5		| 73.1		|
> > >
> > > **TLP**
> > >
> > > | Method		| Task-specific supervision 	 | success	| precision 	|
> > > | --- 			| ---		 		| ---	 	| --- 		|
> > > |KCF			|n				|8.4		|6.3		|
> > > |ECO			|n				|20.2		|21.2		|
> > > |SiamFC		|y				|23.5		|28.4		|
> > > |UniTrack [imagenet features] |n			|25.4		|23.2		|

---

### Official Review · Reviewer_CNPB · 2021-07-16

**Rating:** 7
**Confidence:** 4

**Summary:**

This paper proposes using a unified self-supervised framework for various video-related tracking tasks including single-object bounding box (SOT) or mask propagation (VOS) or multiple object bounding box (MOT), mask (MOTS) or keypoints (PoseTrack) association of detections across video frames. The idea proposed in this paper is inspired by the finding that mid-level representations of networks trained either with full-supervision (e.g., for object classification) or via self-supervised learning for a pre-text task are believed to contain information for establishing dense correspondence between objects and instances. The authors propose to use a single backbone pre-trained network to extract fc3/fc4 features for these various tasks and further employ methods based on dense affinity/ discriminative correlation filters/Hungarian matrix, etc to perform the various propagation or association tasks. The main finding that the authors share is that a common task-agnostic mid-level representation learned using ImageNet is competitive with many supervised and unsupervised task-specific algorithms designed for individual tasks.

**Limitations And Societal Impact:**

The authors addressed the limitations of their work in the broader societal context.

**Main Review:**

Originality: The paper is original in several aspects: -- (a) in investigating a large number (5) of different but closely related tracking tasks together in a unified manner, (b) in highlighting the competitiveness of a single representation learned using ImageNet for many video-related tasks and proposing ways to use them effectively for the different tasks, (c) in analyzing for the first time as to how well self-supervised representations work at these tasks and what their shortcomings are. In summary, this paper proposes a new task, a new research paradigm and a new algorithm.

Quality: Overall the method is technically sound, well-motivated and the experiments are thorough. However, I note some concerns below.

1. With regards to unsupervised learning approaches for association-type tasks there exists the following recently published work, where the authors train a unified network for MOTS and PoseTrack: "Fu et al., Learning to Track Instances without Video Annotations, CVPR 2021". It is not cited. It would be good to cite this work in the reference section and compare against it in Table 1(e) of the main paper and Table 5 of the supplementary material.

2. How critical is the modification of the stride of the original ImageNet18 to the performance of UniTrack? Did the authors also apply this modification to all the networks of the competing methods that they compared against? If not, for a fairer comparison, could the authors also provide the equivalent performance numbers for UniTrack (in Table 1), but without this stride modification?

Clarity: The paper is quite clear and easy to understand. The supplementary document is extensive and provides detailed implementation steps. A few typos in the main document are noted below:

- ln 26: "form" should be "forming"
- ln 27: "differ" should be "differing"
- ln 207: "calculate" should be "calculating"
- ln 263: "/" should be "."

Significance: This work is likely to have a broad impact across the computer vision research community and is likely to be of interest to a wide audience. The fact that it proposes to unify networks for many different, but related video tracking tasks is also quite nice and influential from the point of view of practitioners of computer vision technology as well. It encourages simplification of network design and hence reduction in inference compute cycles, which is an important consideration in real-world applications.

**Time Spent Reviewing:**

4

---

> ### Author Response · Authors · 2021-08-10
> **Response to reviewer CNPB**
>
>
> We thank reviewer CNPB for their time and feedback, and for recognising the originality and significance of our work. Below we address the concerns expressed.
>
> ----
> **1.**
> > [Comparison with  "Fu et al., Learning to Track Instances without Video Annotations, CVPR 2021"].
>
> Thank you for pointing out this relevant reference. We extend Table 1e (main paper) and Table 5 (supplementary) below to compare against it (reported as TWVA). We will also update the paper to reflect these changes.
>
> **PoseTrack-2018 val split**
>
> | Method    | Task-specific supervision     | IDF-1     | *IDs*    | *MOTA* |
> | ---         | ---         | ---     | ---     | ---     |
> | KeyTrack     |      Yes           |    -          |   -    |  66.6    |
> | LightTrack     |      Yes           |  52.2    |3024    |  64.8    |
> | TWVA     |       No           |   -         |   -    |  64.7    |
> | UniTrack [vanilla-imagenet]    |       No           | 73.2    | 6760    |  63.5    |
>
> **Youtube-VIS val split**
>
> | Method        |Task-specific supervision     | Track mAP    |
> | ---             | ---         | ---         |
> | OSMN         | yes         |    27.5    |
> | DeepSORT        | yes        |    26.1    |
> | MaskTrackRCNN     | yes         |    30.3    |
> | SipMask         | yes         |    32.5    |
> | TWVA         | no         |   32.9        |
> | UniTrack [vanilla-imagenet]        | no        |   30.3        |
>
> From the results we can see how TWVA is modestly superior to UniTrack for the problems of PoseTrack and VIS (which we consider as a sixth scenario in the supplementary material). This is not surprising though, as TWVA is trained on similar datasets to the ones used for testing; UniTrack, instead, is trained on a completely different distribution (ImageNet).
>
> ----
> **2.**
> > _“How critical is the modification of the stride [...] to the performance of UniTrack?”_
>
> In SOT and VOS tasks, it is common practice to modify the total stride to 8 by decreasing the strides of residual blocks 3 and 4. For the sake of clarity, below we list the architectures and corresponding strides used by methods from Table 1, and provide performance for UniTrack both with stride 8 (used in the paper) and 16. Note that, for association tasks, some methods adopt re-ID models as appearance models. These re-ID models output a single vector for each observation, therefore we denote the total stride before the final pooling or linear layer as the overall stride.
>
> **SOT@OTB-2015**
>
> | Method        | Task-specific supervision     | arch         | stride    | AUC |
> | ---             | ---                 | ---         | ---     | ---     |
> |SiamFC        |y                |AlexNet     | 8    |58.2    |
> |SiamRPN        |y                |AlexNet    |8    |63.7    |
> |SiamRPN++        |y                |ResNet50    |8    |69.6    |
> |UDT            |n                |Conv_x2    |1    |59.4    |
> |LUDT            |n                |Conv_x2    |1    |60.2    |
> |UniTrack_XCORR    |n                |ResNet18    |8    |55.5    |
> |UniTrack_DCF    |n                |ResNet18    |8    |61.8    |
> |UniTrack_XCORR    |n                |ResNet18    |16    |45.5    |
> |UniTrack_DCF    |n                |ResNet18    |16    |48.6    |
>
>
> **VOS@DAVIS-2017 val split**
>
> | Method        | Task-specific supervision     | arch         | stride    | J-mean|
> | ---             | ---                 | ---         | ---     | ---     |
> |SiamMask        |y                |ResNet50    |8    |54.3    |
> |FEELVOS        |y                |Xception-65    |4    |63.7    |
> |STM            |y                |ResNet50    |8    |79.2    |
> |Colorization        |n                |ResNet18    |8    |34.6    |
> |TimeCycle        |n                |ResNet50    |8    |40.1    |
> |UVC            |n                |ResNet18    |8    |56.7    |
> |CRW            |n                |ResNet18    |8    |64.8    |
> |VFS            |n                |ResNet50    |16    |66.5    |
> |UniTrack        |n                |ResNet50    |8    |58.4    |
> |UniTrack        |n                |ResNet50    |16    |51.0    |
>
>
> **MOT@MOT-16 test set**
>
> | Method        | Task-specific supervision     | arch         | stride    | IDF-1|
> | ---             | ---                 | ---         | ---     | ---     |
> |POI            |y                |Inception    |16    |65.1    |
> |DeepSORT-2        |y                |WRN        |8    |62.2    |
> |JDE            |y                |DarkNet-53    |8    |55.8    |
> |CTracker        |y                |ResNet50    |Multi-scale|57.2|
> |TubeTK        |y                |3D-ResNet    |Multi-scale|62.2|
> |TraDes        |y                |DLA-34    |4    |64.7    |
> |CSTrack        |y                |DarkNet-53    |8    |71.8    |
> |FairMOT        |y                |DLA-34    |4    |72.8    |
> |UniTrack        |n                |ResNet50    |8    |71.8    |
> |UniTrack        |n                |ResNet50    |16    |72.0    |
>
>
>
> **MOTS@MOTS Challenge val split**
>
> | Method        | Task-specific supervision     | arch         | stride    | IDF1    |
> | ---             | ---                 | ---         | ---     | ---     |
> |TrackRCNN        |y                |ResNet101    |32    |42.4    |
> |SORTS        |-                |-        |-    |57.3    |
> |PointTrack        |y                |-        |-    |42.9    |
> |GMPHD         |n                |-        |-    |65.6    |
> |COSTA        |-                |-        |-    |70.3    |
> |UniTrack        |n                |ResNet50    |8    |67.2    |
> |UniTrack        |n                |ResNet50    |16    |66.5    |
>
>
> **Pose Tracking@PoseTrack-2018 val split**
>
> | Method        | Task-specific supervision     | arch         | stride    | IDF1    |
> | ---             | ---                 | ---         | ---     | ---     |
> |MDPN            |y                |flow        |-    |-    |
> |OpenSVAI        |y                |flow        |-    |-    |
> |Miracle        |-                |-        |-    |-    |
> |KeyTrack        |n                |IOU        |-    |-    |
> |LigthTrack        |y                |IOU+GCN    |-    |52.2    |
> |UniTrack        |n                |ResNet18    |8    |73.2    |
> |UniTrack        |n                |ResNet18    |16    |72.8    |
>
>
> From the results we find that modifying the total stride from 16 to 8 indeed brings significant improvements in propagation tasks (SOT, VOS) and only marginal in association tasks. In fact, stride 16 is actually slightly better for MOT.
> Thanks for pointing this analysis, we will update the paper to add these numbers and comment on the importance of stride value for propagation tasks.
>
> -----
> **3.**
> > _“ few typos in the main document”_
>
> Thanks for pointing these out, we will fix the paper accordingly.
>
> ----
> **4.**
> As a final note: we have tested our framework with the appearance model of the very recent self-supervised method (VFS), from the paper “Rethinking Self-Supervised Correspondence Learning: A Video Frame-level Similarity Perspective" [Xu&Wang; ICCV 2021]). We found that it outperforms SimCLRv2 (the best self-supervised model from our submission) on every metric and task except one. For the results, please see the general comment on the top of this page titled “UniTrack results with VFS features”.

---

> > ### Comment · Reviewer_CNPB · 2021-09-01
> > **Response to Author(s)**
> >
> > I thank the authors for their detailed answers to all my questions. I am satisfied with their responses and my doubts about the network stride modification being consistent for most networks have been clarified.
> >
> > Overall, I retain my positive feedback for this paper after the rebuttal as well and will keep my original rating. I recommend accepting this paper.

---

### Author Response · Authors · 2021-08-10
**UniTrack results with VFS features**

We tested our framework UniTrack with the very recently introduced VFS approach from the paper _"Rethinking Self-Supervised Correspondence Learning: A Video Frame-level Similarity Perspective"_ (Xu&Wang), which will be presented at ICCV 2021.
Without any modification or hyperparameter adaptation, using their appearance model improved over the performance of the previously best-performing self-supervised model we tested: SimCLRv2. A condensed view of the results is shown below:

||SOT:AUC_xcorr|SOT:AUC_DCF|VOS:Jmean|MOT:IDF|MOT:HOTA|MOTS:IDF|MOTS:HOTA|PoseTrack:IDF|PoseTrack:HOTA|
|---            | ---    | ---    |---    |---    |---    |---    |---    |---    |---    |
|UniTrack [SimCLRv2]|50.0    |**61.7**    |61.6    |67.6    |58.1    |69.1    |70.4    |72.5    |7228    |
|UniTrack [VFS]    |**51.1**    |60.3    |**62.8**    |**74.1**    |**62.6**    |**71.0**    |**72.1**    |**73.3**    |**6731**    |

---

### Decision · Program_Chairs · 2021-09-27

**Decision:**

Accept (Poster)

**Comment:**

Initially the paper received mixed positive review: three accept (7) and 1 reject (3).  The reviewers appreciated the originality (unified framework for 5 tracking tasks, new similarity measurements and association algorithm, evaluation of supervised and self-supervised representations on 5 tracking tasks).  The reviewers' main concerns were:
1. additional results with new unsupervised methods.
2. some writing/clarity issues
3. didn't use multi-task learning to train the framework.
4. comparison on SOT uses out-of-date dataset and older methods. need further evaluation on more recent datasets.
5. for tracking model, appearance model needs to be specialized for the current environment, rather than a unified appearance model.
6. missing theoretical analysis that appearance models can cover most changes in target appearance.
7. possible unfair comparisons since some methods not pre-trained on imagenet.

In the response, authors provided more experiment results (point 4: SOT results on 5 larger datasets; point 1: another SSL method) and further clarifications and explanations (points 2, 3, 5, 6, 7). The 3 positive reviewers were satisfied with the response, while the negative reviewer was only partially satisfied and raised their rating to "4".  The remaining concern of the negative reviewer was mainly about the theoretical analysis (point 6).  The AC tends to discount this concern, since theoretical analysis seems outside of the scope of this paper, given the large amount of empirical results. The AC agrees with the 3 positive reviewers that the paper provides an interesting new way to unify tracking tasks, and to evaluate representation learning methods on tracking tasks. Thus the AC recommends accept. The authors should update the paper according to the review comments and discussion.